# ⚘Pollinator: Optimal Matchmaking in an Intelligence Marketplace

## Abstract

The rapid growth of the intelligence marketplace has created an abundance of Large Language Model (LLM) producers, each with different cost–performance tradeoffs, making optimal selection challenging and resource-intensive. We present Pollinator, a novel router that integrates a frugal, data-efficient predictor with an online dual-based optimizer. The predictor combines graph-based semi-supervised learning with an Item Response Theory (IRT) head, reducing training cost by up to $49\%$ while improving predictive accuracy over prior state-of-the-art. The optimizer formulates matchmaking as a strongly convex problem, which allows efficient dual-to-primal conversion for real-time serving. Extensive experiments demonstrate that Pollinator delivers superior cost–performance tradeoffs: achieving $0.43\%$-$1.5\%$ gains at $71\%$-$93\%$ of the cost of state-of-the-art router, $3$-$5\%$ gains at only $1.9$-$3\%$ of the cost of the best individual producer, and up to $10.6\%$ higher accuracy at just $0.3$-$35.7\%$ of the cost on challenging real-world benchmarks such as BFCL-V3 and MMLU-Pro. Finally, the interpretability of learned query difficulties and model abilities demonstrates Pollinator's effectiveness for dynamic and cost-efficient intelligence matchmaking.

## 1 Introduction

**The Intelligence Marketplace.** The commoditization of (artificial) intelligence, which gave birth to the phrase *Intelligence on Tap*, coupled with the rapid proliferation in applications that exploit it as a *Design Material* Holmquist (2017), catalyzed a Cambrian explosion of intelligent applications. The author of such an application, however, faces a problem of plenty: there are many producers of intelligence[1] with varying cost-performance tradeoffs on generic benchmarks, making it hard to choose an optimal one appropriate specific to the application at hand. Furthermore, given the frequent updates that alter performance and rapidly falling costs, the process of optimization has to be repeated continuously – thus putting a constant demand on the author's resources.

**Matchmaking with Router.** To remedy, *router* – which routes each request independently to a producer from a pool based on projected cost-performance tradeoff – was conceptualized Hu et al. (2024). A canonical router consists of two components: a collection of *predictors* that project the cost and performance for each request-producer pair; and, an *optimizer* that makes the tradeoff based on the projection.

**Predictor & Data Efficiency.** While a wide gamut of predictors, ranging from simple $k$-Nearest Neighbor Hu et al. (2024) to sophisticated Small Language Model with bespoke Bradley-Terry head Ong et al. (2024) and Item Response Theory Song et al. (2025), have been explored, the angle of data-efficiency has remained underexplored. Recently, Tsiourvas et al. (2025) approached data-efficiency in predictor through the lens of causal inference. However, we remark that the causal-inference setting is overly restrictive: one can indeed send the same request (unit) to more than one producers (treatments) in order to gauge cost and performance (treatment effects), which is a departure from the assumptions in causal-inference. In this work, we combine the superior predictive performance rendered by Item Response Theory grounded in psychometry, with the data-efficiency afforded by graph-based semi-supervised learning to design Pollinator, a novel data-efficient predictor.

---

[1] A search in *Hugging Face* for Transformer-based models with 3B+ parameters results in 170K+ hits. The benchmarking service, *Artificial Analysis*, indexes 250 frontier models.

**Optimizer & Online Matchmaking.**   The prevalent approach to matchmaking in the literature has been to compute the *utility* of each producer for the given request by blending the predicted performance and cost either with a affine combination (with *willingness-to-pay* as a hyper-parameter) Hu et al. (2024), or with a convex combination (again a hyper-parameter) Song et al. (2025). Along similar lines, Somerstep et al. (2025) advances the frontier by designing rate-optimal predictors for cost and performance. Mei et al. (2025) frames the problem as a Linear Program, and recovers the primal solution from the dual variables. Taking inspiration from an optimizer designed for two-sided marketplace Agarwal et al. (2012), we choose to frame the optimization problem as a strongly convex program, which simplifies the dual-to-primal conversion to facilitate online serving. See Appendix B for additional related works.

**Contributions.**   In summary, we make the following contributions: ❶ POLLINATOR implements its predictors atop Graph Convolutional Network Kipf (2016) with a novel Item Response Theory (IRT)-based head that cuts down the training cost by $49\%$, while surpassing the predictive performance of the vanilla IRT-based predictor proposed in Song et al. (2025); ❷ the (Lagrangian) dual-based online serving scheme, coupled with the strongly convex primal program delivers superior cost-performance tradeoff than the Linear Program-based serving scheme Song et al. (2025) by boosting performance by $0.43\%$-$1.5\%$ at $71\%$-$93\%$ of the cost; ❸ in the in-domain and out-of-domain setting proposed in Song et al. (2025), POLLINATOR yields 3-5% boost in performance at mere $1.9$-$3\%$ of the cost of the best producer; ❹ report superior performance in two novel settings on real-world and contemporary benchmarks, such as BFCL-V3 (tool-calling) and MMLU-Pro, where POLLINATOR achieves 3-10.6% higher accuracy at only 0.3-35.7% of the best producer's cost.

## 2   INTELLIGENCE MARKETPLACE & THE MATCHMAKING PROBLEM

**Background.**   The *matchmaker* dispatches each inference request emanating from the consumer, enumerated with $i \in [M]$, in a just-in-time fashion to a request-specific optimal producer, $j \in [N]$. Upon receiving the response, we assume that the matchmaker possesses the ability to compute a *ex post* quality – such as accuracy, $a_{ij} \in \mathbb{R}_+$ – as well as the resource consumption – such as cost, $c_{ij} \in \mathbb{R}_+$ and latency $t_{ij} \in \mathbb{R}_+$ – metrics. Furthermore, we assume that the matchmaker possesses the corresponding *ex ante* estimates, $\hat{a}_{ij}$, $\hat{c}_{ij}$ and $\hat{t}_{ij}$, *before dispatching the request*. Equipped with the ex ante estimates, informally, the role of the matchmaker is to maximize the sum total of response quality, while obeying guardrails on total inference cost, and possibly other resource consumption metrics. We now frame the matchmaker's optimization problem formally.

**Optimal Matchmaking.**   At this point, we distinguish between two settings. First, *batch inference*, where all the inference requests, $i \in [M]$, are known *a priori*. This formalizes the setting a certain class of consumers operate in, e.g., document summarization and information extraction. We posit the matchmaker's objective as to maximize the total ex ante response quality, $\sum_{i\in[M]} \sum_{j\in[N]} x_{ij}\hat{a}_{ij}$, while obeying a guardrail on total inference cost, $\sum_{i\in[M]} \sum_{j\in[N]} x_{ij}\hat{c}_{ij} \leq C$, where $x_{ij}$ is the collection of *primal* variables lying on the probability simplex, $\Delta^N$, defined with the constraints $x_{ij} \geq 0, \forall i \in [M], \forall j \in [N]$ and $\sum_{j\in[N]} x_{ij} = 1, \forall i \in [M]$. We remark that the consumer may want to impose additional constraints, such as minimum volume commitments, where each producer is guaranteed to receive a specified minimum volume of inference request, $\sum_{i\in[M]} x_{ij} \geq M_j, \forall j \in [N]$. The second setting is *online inference*, where the inference requests arrive in a stream, along with the corresponding ex ante estimates. Specifically, at time $i$, the decision $\mathbf{x}_i \in \Delta^N$ has to be taken, *without a foreknowledge of the upcoming requests*, $\mathbf{x}_k, \forall k > i$. The ex post cost and quality are defined, in this case, over a long horizon, $M$. [2] Lastly, the matchmaker will be required to follow a reference policy, $q_{ij}$, where the desired level of proximity is expressed as $\frac{1}{2}\gamma \sum_i \sum_j (x_{ij} - q_{ij})^2$. In order to ease the exposition, we now focus on the setting where quality and the cost are the only constraints at play. We remark that our framework extends to a more general setting, and can incorporate additional linear constraints, such as minimum volume commitments and $p95$ latency. Thus, the canonical form the matchmaker's optimization problem assumes can be expressed as follows.

---

[2]We remark that in practice, certain additional guardrails become desirable in the online setting: e.g., on $p95$ latency – which can also be specified as a linear constraint, in terms of Conditional Value-at-Risk (CVaR).

**Definition 1** (Optimal Matchmaking)**.**

$$\min_{\mathbf{x} \in \Delta^N} \frac{1}{2}\gamma\|\mathbf{x} - \mathbf{q}\|_F - \frac{1}{M}\sum_{i=1}^{M}\sum_{j=1}^{N} x_{ij}\hat{a}_{ij} \text{ s.t. } \sum_{i=1}^{M}\sum_{j=1}^{N} x_{ij}\hat{c}_{ij} \leq C \tag{1}$$

Note that Eq. 1 succinctly describes the online version of the problem as well, assuming $C$ is the long-horizon budget applicable over a span of $M$ requests. Before turning our attention to the solution of Eq. 1, we lay down the design desiderata for the POLLINATOR – the novel optimal matchmaker presented in the present work.

**Desiderata.** In order to ensure practicality of the POLLINATOR system, we impose 2 design desiderata: ❶ *Frugality.* The cost savings yielded by the matchmaker during inference must not be offset by the cost of training its predictors; ❷ *Safety.* For a consumer to be able to relinquish its control over the choice of the producer, it must be assured adherence to the specified guardrails. We now detail how these desiderata guide the design of POLLINATOR.

## 2.1 GRAPH-BASED SEMI-SUPERVISED LEARNING & ENSURING FRUGALITY

We design a two-tower architecture reminiscent of recommendation system: the first tower encodes the request emanating from the consumer; the second tower encodes the producers; while a combiner combines the output of the two towers and emits the ex ante estimates. We detail each of the components below.

**Request & Producer Towers.** The prompt in request $i$, $p_i \in \Sigma^*$, is first encoded into a dense vector, $x_i := \text{Enc}^{\text{R}}(p_i)$, where $\text{Enc}^{\text{R}} : \Sigma^* \mapsto \mathbb{R}^d$ is a pre-trained BERT encoder on vocabulary $\Sigma$, that extracts the embedding corresponding to the [CLS] token that is appended to the prompt. On the set of prompts in the training dataset, we then induce a $k$-nearest neighbour graph, $\mathcal{G}(\mathcal{V}, \mathcal{E})$, where the similarity between nodes $v_i, v_j \in \mathcal{V}$ is defined in terms of the cosine similarity of their corresponding embeddings, $\cos(\angle x_i x_j)$. Let $A$ denote the adjacency matrix of this graph, and let $\tilde{A} := A + I$ denote the adjacency matrix of $\mathcal{G}$, with self-loops added. On this graph, the request tower implements a Graph Convolutional Network (GCN), which propagates information between two successive layers, $l$ and $l + 1$, with $H^{(l+1)} = \sigma(\tilde{D}^{-\frac{1}{2}}\tilde{A}\tilde{D}^{-\frac{1}{2}}H^{(l)}W^{(l)})$. The embedding of node $v_i$ at layer $l = 0$ is set to its embedding, $H_i^{(0)} = x_i$. Lastly, the embedding of the node $v_i$ from the last layer of the GCN, $H_i^{(L)}$, is mapped to a vector, $\alpha_i \in \mathbb{R}^D$, and a scalar, $\beta_i \in \mathbb{R}$, via two learnable linear projections, $W_\alpha, W_\beta$, respectively. Similarly, each producer, $j \in [N]$, is mapped to a embedding, $\theta_j \in \mathbb{R}^D$, with an encoder, $\text{Enc}^{\text{P}} : [N] \mapsto \mathbb{R}^D$. In our experiments, the encoder is a pre-trained BERT that encodes the textual description of the producer, or a simple lookup-based learnable linear projection, $W_\theta$.

**Combiner.** Inspired by Item Response Theory (IRT), the combiner treats $\alpha_i$ as the *discrimination* parameter, which intuitively models the skill-set required to generate a high-quality response to request, $i$, and treats $\beta_i$ as its *difficulty*. On the other hand, $\theta_j$ models the skill-set offered by the producer $j$. IRT posits that the probability of obtaining a high-quality response improves with the degree of match between the skill-set required to process request $i$, and those offered by producer $j$ – and is modulated by the difficulty of the request. This intuition is operationalized as: $\mathbb{P}\{Y_{ij} = 1 := \sigma(\alpha_i^T \theta_j - \beta_i)\}$, where $\sigma(x) = \frac{1}{1+e^{-x}}$ is the usual sigmoid function.

**Training & Inference.** During training, the performance predictor is fit by minimizing the binary cross-entropy loss. The cost estimate is simply taken to be the average cost in training dataset. During inference, we first induce a graph among the incoming request and its $k$ nearest neighbors in the training dataset, and then run the forward-pass for both the towers. Figure 7 in Appendix A.10 summarizes the salient workflows.

---

**Algorithm 1** Dual Serving Scheme

---

1: **Input:** Request $i$; Dual $\lambda$.
2: **Output:** Primal serving scheme $\{x_{ij}\}_{j=1}^N$.
3: Fetch ex ante predictions $\{\hat{a}_{ij}\}_{j=1}^N$ and $\{\hat{c}_{ij}\}_{j=1}^N$               ▷ Invoke predictors.
4: Compute utilities, $\{u_{ij}\}_{j=1}^N$, and sort them into $u_{i(1)}, \cdots, u_{i(N)}$     ▷ Compute and sort utilities.
5: $U = \gamma, j = 1$
6: **repeat**
7:     **if** $u_{i(j)} + \frac{U - u_{i(j)}}{j} \leq 0$ **then**
8:         $j = j - 1$; **break**
9:     **else**
10:         $U = U - u_{i(j)}$; $j = j + 1$; **continue**
11:     **end if**
12: **until** $j \geq N$
13: $\nu_i = \frac{U}{j}$
14: $x_{u(k)} = \frac{u_{i(k)} + \nu_i}{\gamma}, \forall k \leq j$; $x_{u(k)} = 0, \forall k > j$

---

## 2.2 Dual-based Optimization & Adherence to Safety

The primal optimization problem formulated in Eq. 1 comprises a strongly-convex objective and several linear constraints, thus rendering several off-the-shelf solvers immediately applicable at a first glance. However, in practice, its deployment faces two challenges.

**Predict-and-Optimize.** The first problem arises because of the predict-and-optimize paradigm. Ideally, the primal problem in Eq. 1 should have been defined in terms of the ex post coefficients, $a_{ij}$ and $c_{ij}$. However, we only have access to the corresponding ex ante coefficients, $\hat{a}_{ij}$ and $\hat{c}_{ij}$. Thus issues such as predictor's inaccuracy and mis-calibration plague the constrained optimization via both constraints and the objective. We address this issue via improving the accuracy and the calibration of the predictors, and leave a more principled investigation based on the predict-then-optimize framework for a future work.

**Online Optimization.** The second problem appears in the case of online inference. Without *a priori* knowledge of the $M$ requests, the primal problem cannot even be formulated. POLLINATOR solves it via resorting to the *dual*. The Lagrangian of Eq. 1 is presented in Eq. 2.

$$\min_{\mathbf{x} \in \Delta^N} \max_{\lambda, \delta \geq 0, \nu} \frac{1}{2} \gamma \|\mathbf{x} - \mathbf{q}\|_F - \frac{1}{M} \sum_{i=1}^M \sum_{j=1}^N x_{ij} \hat{a}_{ij} + \sum_{i=1}^M \sum_{j=1}^N \lambda(x_{ij} \hat{c}_{ij} - C) - \sum_i \nu_i (\sum_j x_{ij} - 1) - \sum_i \sum_j \delta_{ij} x_{ij} \tag{2}$$

The *stationarity* condition amongst the Karush-Kuhn-Tucker (KKT) conditions allow us to express the primal solution in terms of the dual variables as, $x_{ij} = \frac{u_{ij} + \nu_i + \delta_{ij}}{\gamma}$, where the *utility*, $u_{ij} := \gamma q_{ij} + \frac{1}{M} \hat{a}_{ij} - \lambda \hat{c}_{ij}$. However, this still does not yield a serving plan, given the presence of request-dependent dual variables, $\nu_i$ and $\delta_{ij}$, in the numerator.

In order to get rid of the request-dependent dual variables, we need to appeal to the *complementary slackness* condition amongst the KKT conditions, which leads to the following proposition.

**Proposition 1** (Utility)**.** *In the optimal solution for request $i$, assume producer $j_1$ has more utility than producer $j_2$, $u_{ij_1} \geq u_{ij_2}$. If $x_{ij_2} > 0$, then $x_{ij_1} \geq x_{ij_2} > 0$.*

When the request $i$ arrives, armed with the ex ante estimates, $\hat{a}_{ij}$ and $\hat{c}_{ij}$, and the request-independent dual variable, $\lambda$, we first rank the producers as per their respective utilities. Let $u_{i(1)}, \cdots, u_{i(N)}$ represent the ranked list, where $u_{i(1)} \geq \cdots \geq u_{i(j^*)} \geq \cdots \geq u_{i(N)}$. Proposition 1 illuminates a structure in the solution: $x_{i(1)} \geq \cdots \geq x_{i(j^*)} \geq \cdots = x_{i(N)}$, where $x_{i(j^*)}$ is the smallest non-zero element. Thus, complementary slackness ensures $\delta_{i(j)} = 0$, when $j \leq j^*$, and, primal feasibility ensures, $\sum_{j=1}^{j^*} x_{i(j)} = 1$. This insight allows us to eliminate the last remaining request-specific dual variable, $\nu_i$, from the expression for the primal solution $x_{ij}$, by ensuring $\nu_i = \frac{\gamma - \sum_{j=1}^{j^*} u_{i(j)}}{j^*}$. Proposition 2 distils this insight into a operational definition of $j^*$, which culminates into the optimal primal serving scheme presented in Algorithm 1. Algorithm 1 is dominated

by the sorting step (Line 4), giving an overall complexity of $\mathcal{O}(N \log N)$ for a model pool of size $N$ The detailed runtime and complexity analysis is provided in Appendix A.8.

**Proposition 2** (Optimal Stopping). *$j^*$ is the maximum $j$ such that $u_{i(j)} + \frac{\gamma - \sum_{k \leq j} u_{i(k)}}{j} > 0$.*

Note that LLM pricing depends on multiple deployment factors—cloud provider, instance type, region, reservation model, negotiated rates, and API-specific pricing (e.g., per-million-token rates or provisioned throughput). In this work, we abstract these nuances by assuming a per-million-token rate card (separately for input, intermediate, and output tokens), derived from the underlying unit economics of the deployment. Whenever this rate card changes, the optimization problem (Eq. 1) must be re-solved and the resulting dual variables redistributed to the workers executing Algorithm 1. Proposition 1-2 and Algorithm 1 extend seamlessly to any optimization problem with strongly convex objective and linear constraints (similar to Eq. 1) – only requiring a change in the expression for utility, $u_{ij}$ (see Sec 2.2). Every new constraint (e.g minimum LLM usage volume commitment) added to Eq. 1 need to be added to $u_{ij}$ with an appropriate sign (depending on whether the constraint is covering- or packing-type) and multiplied by its own Lagrangian dual variable.

## 3 EXPERIMENTAL SETUP

### 3.1 BENCHMARK & IMPLEMENTATION

**In-Domain Dataset (ID).** Following Song et al. (2025), we evaluate POLLINATOR's in-domain generalization capability over 8 datasets: ① **MMLU** (Hendrycks et al., 2020) (reasoning and knowledge across 57 domains), ② **CMMLU** (Li et al., 2024) (a Chinese incarnation of MMLU), ③ **ACLUE** (Zhang & Li, 2023) (ancient Chinese language-understanding), ④ **ARC_C** (Clark et al., 2018) (advanced reasoning), ⑤ **HotpotQA** (Yang et al., 2018) (question-answer requiring multi-hop reasoning), ⑥ **SQuAD** (Rajpurkar et al., 2018) (reading comprehension), ⑦ **MATH** (Hendrycks et al., 2021) (competition-level mathematics), and, ⑧ **MBPP** (Austin et al., 2021) (basic coding). 20 LLMs were in the candidate pool (see Table 17 in Appendix A.4). POLLINATOR is trained on 70% of the *combined* dataset, and tested in-domain on the remaining 30%.

**Out-of-Domain Dataset (OOD).** Following Song et al. (2025), we evaluate POLLINATOR's out-of-domain generalization capability over 4 datasets that we not part of the training: ⑨ **CEVAL** (Huang et al., 2024) (tasks in Chinese language), ⑩ **CommonsenseQA** (Talmor et al., 2019) (commonsense reasoning), ⑪ **GSM8K** (Cobbe et al., 2021) (grade-school mathematics), and, ⑫ **HumanEval** (Chen et al., 2021) (coding). The same candidate LLM pool as In-Domain Dataset (**ID**) is employed. We emphasize that 100% of the combined datasets are used in test.

**MMLU-Pro & BFCL-V3.** In order to further evaluate POLLINATOR's efficacy on contemporary datasets and real-world scenarios such as tool-call, we benchmark in-domain on 2 additional datasets: ⑬ **MMLU-Pro** (Wang et al., 2024) (extends **MMLU** with harder multiple-choice questions with 10 possible choices, instead of 4), and, ⑭ **BFCL-V3 (Simple)** (Patil et al., 2025) (reasoning and ability to call external tools and APIs in real-world setting – a key skill for agentic applications). The LLM candidate pool consists of 15 members, including GPT and Gemini families, for **MMLU-Pro** (details in Table 18 in Appendix A.6), and 10 for **BFCL-V3 (Simple)**, including OpenAI o-series and Llama-3.1 families (see Table 19 in Appendix A.6 for an exhaustive list). For each of these 2 datasets, 70% is used for training and the rest 30% for testing (we emphasize that we do not combine the datasets). Table 12 in Appendix A.3 enumerates comprehensive details on all 14 datasets, including evaluation metrics and dataset cardinality.

**Implementation.** We use `bert-base-uncased` [3] as the request encoder, $\text{Enc}^{\text{R}}$. The producer encoder, $\text{Enc}^{\text{P}}$, simply encodes the producer ID, $j \in [M]$. The GCN consists of $L = 2$ layers with hidden dimension of 64 and dropout rate of 0.3.The producer skill vector $\theta_j$ has dimension 16, except for the out-of-domain data, where it has dimension 25. The $k$NN graph construction uses $k = 3$ nearest neighbors, with edge weights (optionally) set to the cosine similarity, $\cos(\angle x_i x_j)$. POLLINATOR is trained with the Adam optimizer (Adam et al., 2014), with learning rate $1 \times 10^{-3}$ and weight decay $1 \times 10^{-5}$ for 200 epochs. During inference, we induce a graph among the test request and its $k = 3$ nearest neighbors. All hyper-parameters were selected based on best performance on a held-out validation set. All experiments are run on 1 NVIDIA A40 GPU with 40GB memory.

---

[3] https://huggingface.co/google-bert/bert-base-uncased

## 3.2 METRICS & BASELINES

**Metrics.** We evaluate routing algorithms by the cost-performance tradeoffs they offer in the test dataset. Given binary ground-truth label, $a_{ij_i} \in \{0, 1\}$, where $j_i$ is the chosen producer for request $i$, **Performance (%)** is defined in a dataset-dependent manner as either of *accuracy*, *F1*, *Exact Match (EM)* and *Pass@1* over the test dataset (see Table 12 in A.3 for dataset-specific evaluation metrics) – a methodology that echoes Song et al. (2025). **Cost ($)** is calculated as per the token counts and the rate-card for the producer $j_i$, where $i$ indexes the requests in test dataset. In order to highlight a few salient points on the cost-quality Pareto, we coin performance-first, balanced and cost-first configurations. In particular, in the performance-first setting, we set a large $C$ in Eq. 1, so that the optimizer has a large room to maximize accuracy (performance). In cost-first, we set $C$ in a stringent manner, thus yielding lower performance. Balanced setting sets $C$ to a medium value.

**Baselines.** We compare POLLINATOR against a set of strong baselines: ① **Small LLM** always routes queries to the specifically chosen small model (likely based on number of parameters) for each dataset. Following Song et al. (2025), we use *Ministral-8B-Instruct-2410* as the Small LLM for both the In-Domain and Out-of-Domain datasets. For MMLU-Pro, the Small LLM is Meta-Llama-3.1-70B, while for BFCL-V3(ToolCall), we use GPT-4.1-Nano. Please note the *Small LLM* is **not necessarily the cheapest model**. ② **Large LLM** always routes queries to the largest candidate model. ③ **kNN-Router** is a simple retrieval-based baseline that selects an LLM based on the top-$k$ ($k = 5$ gave the best performance) most similar queries, choosing the lowest-cost model among them. ④ **HybridLLM** (Ding et al., 2024) trains a pre-trained encoder (DeBERTa-v3) with matrix factorization to decide between a small and a large model. ⑤ **RouteLLM** (Ong et al., 2024) uses a binary classifier for pairwise LLM routing. ⑥ **RouterBench** (Hu et al., 2024) provides multiple routing strategies; we adopt its Predictive Router variant to avoid the high cost of querying all candidate models. ⑦ **MIRT-Router** and **NIRT-Router** (Song et al., 2025) are IRT-based methods and serve as our closest baselines. We implement only kNN-Router; results for the others are taken from Song et al. (2025). MIRT-Router is run on MMLU-Pro and BFCL-V3 using their official code, while NIRT-Router is omitted due to its reliance on costly GPT-4o relevance vectors. Following Song et al. (2025), we also generate LLM profile descriptions (Table 22 in Appendix A.11) for candidate models in **MMLU-Pro** and **BFCL-V3 (Simple)**.

Table 1: Comparison of routing methods in In-Domain Dataset across three distinct performance-cost tradeoff scenarios. **Bold** and underline denote the best and second-best results.

| Method | Performance-First | | Balanced | | Cost-First | |
|---|---|---|---|---|---|---|
| | **Performance (%)**↑ | **Cost ($)**↓ | **Performance (%)**↑ | **Cost ($)**↓ | **Performance (%)**↑ | **Cost ($)**↓ |
| Small LLM | 48.70 | **0.31** | 48.70 | **0.31** | 48.70 | 0.31 |
| Large LLM | 77.53 | 12.93 | 77.53 | 12.93 | 77.53 | 12.93 |
| HybridLLM | 54.37 | 1.98 | 52.42 | 1.54 | 56.65 | 2.78 |
| RouteLLM | 77.25 | 12.80 | 73.59 | 11.15 | 66.24 | 7.51 |
| RouterBench | 80.01 | 1.15 | 79.48 | 0.53 | 78.36 | 0.37 |
| kNN-Router | 74.38 | 1.14 | 74.38 | 1.14 | 74.38 | 1.14 |
| MIRT-Router | 80.67 | 0.42 | 80.65 | 0.42 | 80.03 | 0.39 |
| NIRT-Router | 80.69 | 0.55 | 80.41 | 0.43 | 79.37 | 0.41 |
| **POLLINATOR** | **81.38** | 0.39 | **81.38** | 0.39 | **80.09** | **0.26** |

## 4 RESULTS

In **ID** dataset, as seen in Table 1, in the performance-oriented batch inference setting, POLLINATOR delivers $0.8\%$ performance gain over NIRT-Router at $70\%$ of its cost. Similarly, in the cost-oriented setting, POLLINATOR renders $33\%$ cost reduction over MIRT-Router, at a slightly better performance. In the balanced setting, POLLINATOR achieves a $0.9\%$ gain over SOTA at $93\%$ of the cost, and a $5\%$ improvement over the best producer at $3\%$ cost. In **OOD** (Table 2), the relative cost reduction in the balanced setting stands at $28.57\%$, at a similar performance. The $\geq 25\%$ cost advantage holds on MMLU-Pro and BFCL V3 datasets as well with $1.5\%$ and $0.43\%$ gains over SOTA, as seen from Table 3. Overall, POLLINATOR delivers a superior Pareto frontier in the cost-performance plane across ID, OOD, and real-world benchmarks . POLLINATOR effectively routes queries to the most appropriate model, avoiding unnecessary invocation of expensive LLMs while respecting global budget constraints. The whole spectrum of performance and cost across

all datasets is shown in Figure 1 of Appendix 4.1. The oracle results are presented in Table 11 of Appendix A.5. The average performance and associated costs of individual LLMs for each dataset are reported in Tables 13, 14, 15, and 16, corresponding to the In-Domain, Out-of-Domain, MMLU-Pro, and BFCL-V3 (ToolCall) datasets, respectively, in Appendix A.5. The scalability and robustness of POLLINATOR is discussed in detail in Appendix A.1, where we evaluate end-to-end latency, throughput, handling of large model pools, resilience to model and query drift, mitigation of prediction errors and robustness under dynamic pricing and provider availability. These results demonstrate that POLLINATOR maintains high efficiency and reliability under real-world conditions.

Table 2: Comparison of routing methods in Out-of-Domain Dataset.

| Method | Performance-First | | Balanced | | Cost-First | |
|---|---|---|---|---|---|---|
| | Performance (%)↑ | Cost ($)↓ | Performance (%)↑ | Cost ($)↓ | Performance (%)↑ | Cost ($)↓ |
| Small LLM | 59.83 | **0.11** | 59.83 | 0.11 | 59.83 | 0.11 |
| Large LLM | 84.90 | 5.30 | 84.90 | 5.30 | 84.90 | 5.30 |
| HybridLLM | 63.34 | 0.73 | 62.08 | 0.41 | 63.79 | 0.65 |
| RouteLLM | 84.39 | 5.25 | 79.90 | 4.74 | 75.06 | 3.48 |
| RouterBench | 85.50 | 0.26 | 85.75 | 0.16 | 84.62 | 0.12 |
| kNN-Router | 80.92 | 0.29 | 80.92 | 0.29 | 80.92 | 0.29 |
| MIRT-Router | 87.12 | 0.14 | 87.12 | 0.14 | 87.18 | 0.13 |
| NIRT-Router | **87.37** | 0.15 | 87.24 | 0.14 | 87.46 | 0.13 |
| **POLLINATOR** | 87.37 | 0.14 | **87.63** | **0.10** | **87.69** | **0.09** |

Table 3: Comparison of routing methods on MMLU-Pro and BFCL V3 (ToolCall) datasets under the Performance-First setting, reporting model accuracy and associated cost.

| Method | MMLU-Pro | | BFCL-V3 (ToolCall) | |
|---|---|---|---|---|
| | Performance (%)↑ | Cost ($)↓ | Performance (%)↑ | Cost ($)↓ |
| Small LLM[1] | 53.07 | 2.10 | 77.33 | 0.01 |
| Large LLM[2] | 71.61 | 5.01 | 89.33 | 1.85 |
| kNN-Router | 74.23 | 2.55 | 86.67 | 0.02 |
| MIRT-Router | 78.84 | 1.18 | 90.66 | 0.008 |
| **POLLINATOR** | **79.18** | **0.88** | **92.00** | **0.006** |

## 4.1 PERFORMANCE-COST SPECTRUM

To obtain the full performance–cost tradeoff spectrum, POLLINATOR produces multiple operating points by varying the optimizer's hyperparameters, enabling different budget–performance preferences and effectively balancing accuracy against total computational cost. Each scatter plot additionally includes the standalone performance–cost pairs of individual LLMs evaluated under the four settings: In-domain (Figure 1a), Out-of-Domain (Figure 1b), MMLU-Pro (Figure 1c), and BFCL-V3 (ToolCall) (Figure 1d). We also include an *Oracle* point where we choose the best and cheapest LLM for each sample. Comparing these points shows that POLLINATOR consistently achieves more favorable performance–cost tradeoffs, forming a superior efficiency frontier relative to any individual LLM. Because the BFCL-V3 (ToolCall) setting contains a very limited amount of training data, a few standalone LLMs occasionally match or slightly surpass the performance of POLLINATOR. We are confident that with more training data, POLLINATOR would surpass these cases and restore its advantage.

## 5 ABLATION STUDY

We perform a comprehensive ablation study on the performance predictor inside POLLINATOR on **ID** datasets to assess the impact of key design decisions, encompassing: ① request encoder, Enc$^R$; ② choice of $k$ in graph construction; ③ the dimensionality, $D$, of $\theta_j \in \mathbb{R}^D$; ④ fraction of labeled nodes in GCN. Findings are summarized in Table 4 and Table 6. For detailed ablations across all combinations of Enc$^R$, $k$, $D$, and edge weighting strategies, refer to Table 21 in Appendix A.9.

---

[1] Small LLM refers to Meta-Llama-3.1-70B for MMLU-Pro and GPT-4.1-Nano for ToolCall.
[2] Large LLM refers to Gemini-1.5-Pro for MMLU-Pro and o1 for ToolCall.

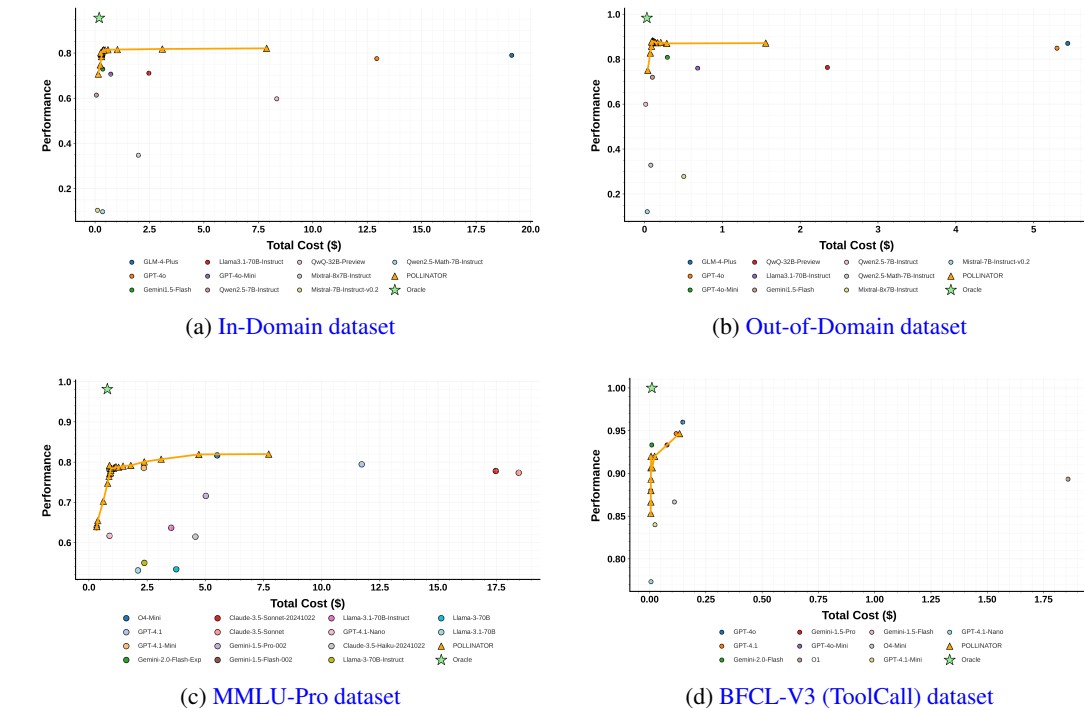

(a) In-Domain dataset

(b) Out-of-Domain dataset

(c) MMLU-Pro dataset

(d) BFCL-V3 (ToolCall) dataset

Figure 1: Performance-cost spectrum across In-Domain, Out-of-Domain, MMLU-Pro, and BFCL-V3 (ToolCall) datasets. Each scatter plot shows performance-cost spectrum of POLLINATOR alongside the standalone performance-cost pairs of individual LLMs and an *Oracle* that selects the best cheapest model per sample.

Table 4: Ablation study of the predictor (without optimizer). Accuracy (%) is reported for different request encoders, graph neighborhood sizes, and varied model ability dimension. POLLINATOR is robust to embedding and neighborhood choices, sensitive to model ability dimension. The best configuration of POLLINATOR is marked with †. Neighborhood size and model ability experiments use *bert-base-uncased* as $\text{Enc}^R$.

| Request Encoder ($\text{Enc}^R$) | | Graph Neighborhood Size ($k$) | | Model Ability Dimension ($D$) | |
|---|---|---|---|---|---|
| $\text{Enc}^R$ | Perf. (% ↓) | $k$ | Perf. (% ↓) | $D$ | Perf. (% ↓) |
| bert-base-uncased† | 82.07 (-) | 3† | 82.07 (-) | 3 | 71.43 (↓12.96) |
| all-MiniLM-L6-v2 | 80.75 (↓1.32) | 5 | 81.92 (↓0.15) | 5 | 80.49 (↓1.58) |
| all-mpnet-base-v2 | 81.70 (↓0.37) | 10 | 81.87 (↓0.20) | 8 | 79.26 (↓2.81) |
| text-embedding-3-large | 81.34 (↓0.73) | 20 | 82.02 (↓0.05) | 16† | 82.07 (-) |
| | | | | 25 | 81.96 (↓0.11) |
| | | | | 35 | 80.52 (↓1.55) |
| | | | | 45 | 80.25 (↓1.82) |

**Request Encoder $\text{Enc}^R$.** We experiment with different $\text{Enc}^R$ to encode requests. The best configuration (POLLINATOR†) achieves 82.07% accuracy. Alternatives such as *all-MiniLM-L6-v2*[4], *all-mpnet-base-v2*[5], and t*ext-embedding-3-large*[6] show a slight drop in performance of 1.32%, 0.37%, and 0.73% respectively, as reported in Table 4 (left), indicating that the predictor is robust to the choice of embedding while preserving strong predictive capability.

**Neighborhood Size ($k$).** We analyze predictor performance under varying graph neighborhood sizes $k$. The best accuracy occurs at $k = 3$ (82.07%), while larger neighborhoods ($k = 5, 10$) reduce accuracy (81.92%, 81.87%), and $k = 20$ only partially recovers it (82.02%). This indicates that large $k$ introduces noisy neighbors, limiting predictor precision (Table 4, middle).

---

[4]https://huggingface.co/sentence-transformers/all-MiniLM-L6-v2
[5]https://huggingface.co/sentence-transformers/all-mpnet-base-v2
[6]https://platform.openai.com/docs/models/text-embedding-3-large

Table 6: Effect of node label masking on predictor performance (relative drop ↓) and training cost (with savings ↑) on the ID dataset, without optimizer.

| Node Masked (%) | Perf. (% ↓) | Training Cost ($) (Saving % ↑) |
|---|---|---|
| 0 | 82.07 (-) | 208.76 (-) |
| 10 | 81.96 (↓0.11) | 194.32 (6.91 ↑) |
| 20 | 82.01 (↓0.06) | 179.54 (13.99 ↑) |
| 30 | 81.97 (↓0.10) | 164.92 (21.00 ↑) |
| 50 | 81.82 (↓0.25) | 135.66 (35.03 ↑) |
| 60 | 81.79 (↓0.28) | 121.10 (42.00 ↑) |
| **70** | **81.48 (↓0.59)** | **106.23 (49.11 ↑)** |

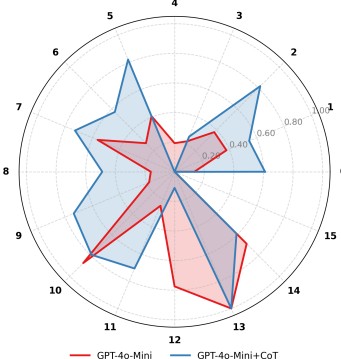

Figure 2: Comparison of GPT-4o-Mini and GPT-4o-Mini+CoT over 16 dimensions. The CoT variant improves performance on most dimensions, showing the benefit of explicit reasoning.

Table 5: Ablation study showing results of POLLINATOR with and without the GCN module across all datasets. Removing the GCN leads to noticeably lower performance and higher cost.

| Method | Datasets | | | | | | | |
|---|---|---|---|---|---|---|---|---|
| | In-Domain | | Out-of-Domain | | MMLU-Pro | | BFCL-V3 (ToolCall) | |
| | Perf (%)↑ | Cost ($)↓ | Perf (%)↑ | Cost ($)↓ | Perf (%)↑ | Cost ($)↓ | Perf (%)↑ | Cost ($)↓ |
| POLLINATOR w/o GCN | 80.20 | 0.63 | 79.18 | **0.13** | 74.23 | 2.55 | 86.67 | 0.02 |
| POLLINATOR | **81.38** | **0.39** | **87.37** | 0.14 | **79.18** | **0.88** | **92.00** | **0.006** |

**Effect of Model Ability Dimension** ($D$). We assess how the model ability dimension $D$ impacts the predictor's performance (Table 4, right), reporting results relative to the optimal configuration ($D = 16$). Extremely low dimensions ($D = 3$) severely underfit, causing a 12.96% drop in accuracy. Moderate dimensions ($D = 5$ or $D = 8$) partially capture model abilities, resulting in 1.58% and 2.81% decreases. Slightly larger dimensions ($D = 25$) perform comparably to the optimum, with only a 0.11% drop, indicating sufficient capacity without over-parameterization. Excessively high dimensions ($D = 35$ or $D = 45$) introduce redundancy, causing 1.55% and 1.82% drops. Intermediate $D$ offers the best balance between expressivity and generalization.

**Impact of the GCN Module.** To evaluate the contribution of the GCN within POLLINATOR, we conduct an ablation in which we entirely remove the GCN module and replace it with a non-learned alternative: simple $k$NN averaging over the neighborhood (i.e., aggregation without message passing). As shown in Table 5, this removal leads to consistent degradation across all datasets. In-domain accuracy drops from 81.38% to 80.20%, out-of-domain from 87.37% to 79.18%, and MMLU-Pro from 79.18% to 74.23%, with corresponding increases in cost. Even on BFCL-V3, accuracy falls (92.00% → 86.67%) and cost rises. These results demonstrate that naive neighbor averaging cannot substitute the learned aggregation performed by the GCN, confirming its essential contribution to POLLINATOR's routing quality.

**Semi-Supervised Cost-Efficient Training.** Labeling all training nodes can be costly, since obtaining responses from commercial LLMs incurs significant expense. However, as GCNs naturally propagate label information across neighboring nodes, full supervision may be unnecessary. To quantify this, we simulate a semi-supervised setting by randomly masking a fraction of training node labels and report results in Table 6. The predictor demonstrates strong resilience to missing supervision. With 70% of nodes masked, the predictor achieves 81.48% accuracy, comparable to the state-of-the-art MIRT-Router w/o optimizer (81.17%, Table 21, Appendix A.9) and slightly below the fully supervised setting (82.07%), while reducing training cost by 49%. Intermediate masking levels (10%–30%) yield proportional savings (6.91%–21%) with minimal performance loss.

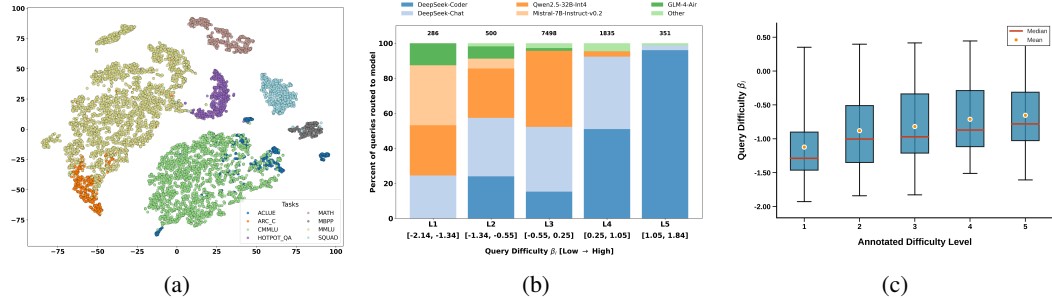

(a)  (b)  (c)

Figure 3: Visualization of interpretability analyses. **(a)** t-SNE projection of learned discrimination vectors $\alpha_i$ across in-domain datasets, showing task-specific clustering. **(b)** Routing distribution of models across query difficulty bins, where lightweight models handle easier queries while advanced models handle harder ones. **(c)** Predicted query difficulties ($\beta_i$) in the MATH dataset, grouped by human-annotated levels, showing monotonic alignment between model estimates and annotations.

### 5.1 Interpretability of Pollinator

**LLM Ability.** We examine ability differences within model families using POLLINATOR. Figure 2 shows GPT-4o-Mini versus its CoT-augmented version, with CoT improving reasoning performance. Similarly, Llama3.1-405B-Instruct outperforms Llama3.1-70B-Instruct (Figure 5, Appendix A.7). Using a model ability dimension $D = 25$, trends align with scaling laws (Kaplan et al., 2020): larger models perform better. Table 13 (Appendix A.5) supports these observations.

**Query Difficulty.** We assess POLLINATOR's ability to estimate query difficulty using the MATH dataset with human-annotated levels. As shown in Figure 3c, the estimated difficulty parameter $\beta_i$ increases monotonically and closely follows the ground-truth progression. Representative examples in Figure 6 (Appendix A.7) further illustrate the strong alignment between POLLINATOR's estimates and human labels.

**Routing Behavior Across Difficulty Levels.** We analyze POLLINATOR routing across queries stratified by difficulty (Figure 3b, $\beta_i$ spans $-2.14$ to $1.84$, divided into L1–L5). While top models like DeepSeek-Coder, DeepSeek-Chat, and GPT-4o achieve highest performance (Table 13 in Appendix A.5), POLLINATOR routes queries cost-efficiently. Easier queries (L1–L2) use lightweight models (Qwen2.5-32B-Int4, Mistral-7B, GLM-4-Air), intermediate bins (L3–L4) show mixed routing, and hardest queries (L5) prefer top-tier models like DeepSeek-Coder.

**Discrimination Vector Analysis.** In POLLINATOR, the discrimination vector $\alpha_i$ encodes the skill requirements for a query. To assess whether these vectors capture task-level structure, we cluster queries based on their learned representations and project them into 2D using t-SNE (Figure 3a). Queries from the same dataset form cohesive clusters, showing that POLLINATOR effectively learns task-aligned skill representations. Some clusters partially overlap, reflecting shared skills: for example, ARC_C and MMLU overlap due to similar reasoning skills, while CMMLU and ACLUE, the only two Chinese datasets, share the embedding space. In contrast, MATH, SQuAD, and MBPP form well-separated clusters, indicating that vectors capture distinct task-specific skill requirements.

## 6 Conclusion

We presented POLLINATOR, a data-efficient and online-serving-capable matchmaker for the intelligence marketplace. POLLINATOR combines a frugal GCN-based predictor with an IRT-head and an efficient dual-optimizer, reducing training cost by up to $49\%$ while outperforming existing state-of-the-art predictors. Extensive experiments on real-world benchmarks, including BFCL-V3 and MMLU-Pro, demonstrate superior cost–performance trade-offs. Furthermore, detailed ablation studies and interpretability validate POLLINATOR's effectiveness for cost-efficient intelligence matchmaking. Future work will extend the framework to incorporate latency and volume constraints and explore adaptive dynamic graphs for evolving requests and producers.

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

## A  APPENDIX

### A.1  SCALABILITY AND ROBUSTNESS OF POLLINATOR

**End-to-End Latency.** To further assess the practical feasibility of our routing framework, we evaluate its end-to-end latency across datasets of varying scales. The bottleneck is the nearest neighbour look-up. Even an exact nearest neighbour search yields sub-100ms p99 (99th percentile) latency, as seen in the Table 7. Note that with approximate nearest neighbour search, especially with HNSW index, the sub-100ms p99 latency can be maintained in industry-scale data (a study[7] from Pinecone reports 50ms p99 at a good recall). It is worth mentioning that the largest dataset reported in the literature on prompt routing has cardinality 1.5M (Feng et al., 2025), which is considered "small" in the parlance of approximate nearest neighbour search.

Table 7: Inference time of POLLINATOR across all datasets. Per-query latency is reported in milliseconds (ms).

| Dataset | Per-Query Inference Time (ms) |
|---|---|
| In-Domain | 69.07 |
| Out-of-Domain | 78.59 |
| MMLU-Pro | 20.37 |
| BFCL-V3 (ToolCall) | 2.66 |

**Throughput**. Industrial vector databases, such as Qdrant, can support 1,200 RPS (Requests Per Second) at 0.99 precision (source[8]). Thus, instead of nearest neighbour lookup, the throughput bottleneck shifts to the web-framework. FastAPI, with its asynchronous async/await primitives, offers high throughput suitable for an industry-grade prompt router – which, moreover, is horizontally scalable. Note that in the current work, we limit our scope to the functional requirements of prompt router, not its non-functional requirements, such as latency/throughput.

**Overhead Analysis.** It's worth noting that the prompt router routes each request to exactly one LLM, either hosted by a provider (e.g., OpenAI, TogetherAI) or self-hosted (e.g., with vLLM) – and in no cases more than one LLM is being invoked. As noted in the latency analysis, prompt router incurs sub-100ms p99 latency, which is negligible, given that self-hosted LLMs take 400-700ms (depends on the parameter count, architecture and the inference engine – and varies across workloads), and those hosted by provider often exceeds $\approx 1.2s$ (even with provisioned throughput, such as PTU in Azure). There are no additional token overhead, as such, beyond those already accounted for under the prompt router latency.

**Robustness w.r.t. Dynamic Pricing.** The rate cards for providers change infrequently. However, constructs such as provisioned throughput (e.g., PTU[9] in Azure Foundry) render the rate card a function of throughput. Even in this case, organizations typically purchase a fixed amount of PTU, rendering the rate card essentially frozen over the contract period (an year). In the (infrequent) event of change in rate card, the dual variables need to be recomputed and deployed via a configuration service to all workers executing Algorithm 1.

**Robustness w.r.t. Availability.** The providers indeed suffer downtimes, and to counteract that, one typically routes to a fallback (which is typically the next available provider in the ordered list, $x_{ij}$ – Line 2 in Algorithm1) after a pre-configured amount of retry. The provision of fallback has been popularized by commercial prompt routers, such as OpenRouter[10].

**Handling Large Model Pool.** In practice, few commercial prompt routers are deployed with model pools of the size 100 (HuggingChat Omni routes[11] across 115 models). More often, they route within the same model family (due to considerations arising from lack of prompt portability – what works best with GPT needn't work with Gemini, as seen in their guides), thus limiting the model

---

[7] https://www.pinecone.io/learn/series/faiss/hnsw/

[8] https://qdrant.tech/benchmarks/

[9] https://learn.microsoft.com/en-us/azure/ai-foundry/openai/concepts/provisioned-throughput?view=foundry-classic&tabs=global-ptum

[10] https://openrouter.ai/docs/features/provider-routing

[11] https://news.ycombinator.com/item?id=45623284

pool size to 10. However, in the hypothetical scenario with 10,000 models (which is practical only when routing across LoRAs, as done by vLLM Semantic Router), we would embrace a two-stage design a la recommender system. The first stage (a high-recall one) would cut down the pool to 100, which the second, high-precision stage will evaluate as per Algorithm 1.

**Total Requests per Second**

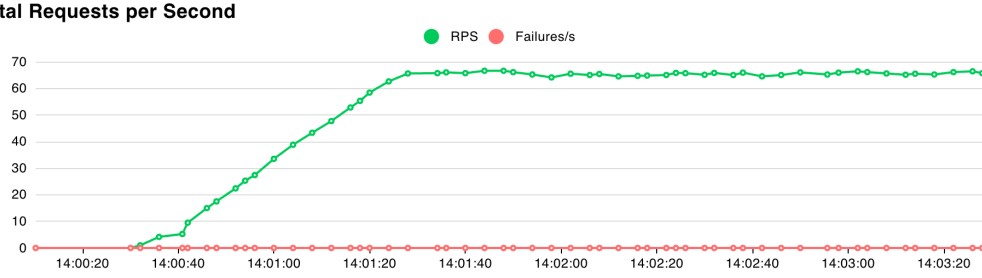

**Response Times (ms)**

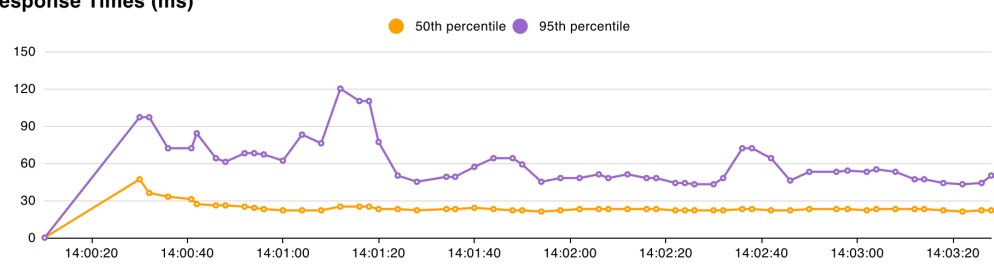

**Number of Users**

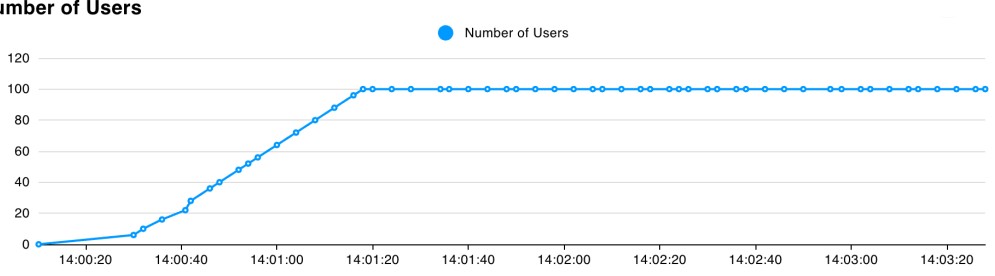

Figure 4: POLLINATOR:Stress Testing Under Increasing Concurrency

**Stress Testing Under Varying Concurrency Levels:** We conducted a realistic load-test with an industry-standard tool, Locust[12], under varying levels of concurrency (at 57 requests-per-second) to evaluate POLLINATOR's inference performance under real-world conditions. The inference engine consists of a) FastAPI[13]-based web-framework, b) Infinity[14] -based embedder, and, c) Qdrant -based approximate k-nearest-neighbour search. All calls to embedder and nearest neighbour searches are non-blocking, via Python's asynchronous coroutine ('async' and 'await') – which mimics the architecture of a typical real-world inference engine. Figure 4 presents the overall system dynamics during this load ramp, including achieved throughput, median (p50) and tail (p95) latencies, and active user concurrency. We additionally report detailed request and response time statistics in Table 8 and Table 9 respectively. The results show that the system sustains stable throughput ( 57 RPS) with zero failures, while maintaining low median latency ($\approx$ 23–28ms) and tightly bounded tail latency (p95$\leq$ 55ms) even under peak concurrency. These observations confirm POLLINATOR 's robust scalability and reliable performance under heavy concurrent workloads.

---

[12]https://docs.locust.io/en/stable/index.html

[13]https://fastapi.tiangolo.com/

[14]https://docs.langchain.com/oss/python/integrations/text_embedding/infinity

Table 8: Aggregate request statistics of POLLINATOR during concurrency stress testing.

| Type | Name | #Req. | #Fails | Avg. (ms) | Min (ms) | Max (ms) | Avg. Size (bytes) | RPS |
|------|------|-------|--------|-----------|----------|----------|-------------------|-----|
| GET | /route | 10,213 | 0 | 27.67 | 12 | 200 | 1182.69 | 56.73 |
| – | Aggregated | 10,213 | 0 | 27.67 | 12 | 200 | 1182.69 | 56.73 |

Table 9: Response time statistics of POLLINATOR during concurrency stress testing.

| Method | Name | 50% | 60% | 70% | 80% | 90% | 95% | 99% | 100% |
|--------|------|-----|-----|-----|-----|-----|-----|-----|------|
| GET | /route | 23 | 25 | 28 | 34 | 43 | 55 | 96 | 200 |
| – | Aggregated | 23 | 25 | 28 | 34 | 43 | 55 | 96 | 200 |

**Handling Drift.** It is indeed true that LLM abilities may drift with subsequent releases of frontier models, or may be caused by iterative fine-tuning of self-hosted models. Similarly, since prompts are frequently updated in deployed systems (e.g., addition of new instructions in prompt, addition/deletion of tools, etc.) – drift can occur. Such drifts are dealt with by re-training the predictors and the (primal) optimizer in POLLINATOR to adapt to the new data distribution. The trigger for re-training (i.e., a drift detection module), however, lies outside POLLINATOR's system boundary to promote simplicity. Alternatively, one can configure a cron-based periodic re-training. However, we note that as shown in Table 2, POLLINATOR outperforms the baselines in Out-of-Domain datasets, thus yielding resilience to drift in prompt distribution out of the box.

Additionally, we conducted a targeted data-drift experiment on the MMLU-Pro dataset, which spans 14 heterogeneous subject areas ranging from law to computer science. At inference, each test query was linked to its $k{=}3$ nearest training neighbors via a kNN graph, and we selected 500 test samples with the lowest average similarity as a proxy for severe distributional drift. POLLINATOR maintained competitive performance (as shown in Table 10) on this subset, demonstrating strong robustness to data drift.

Table 10: Data-drift evaluation of POLLINATOR on MMLU-Pro.

| Dataset | Routing Strategy | Performance (%) ↑ | Cost ($) ↓ |
|---------|------------------|-------------------|------------|
| MMLU-Pro | Performance-first | 79.60 | 0.28 |
| | Balanced | 80.00 | 0.14 |
| | Cost-first | 76.40 | 0.10 |

**Handling Prediction Errors.** While designing POLLINATOR , we acknowledged that performance and cost predictions can be wrong, and since the optimization problem uses them in objective and constraints, it will cause an optimality gap. The present work tackles it by breaking down the long-horizon into epochs, so that at the end of each epoch, the feedback (actual performance yield and cost incurred) from that epoch can be incorporated into the primal optimization problem (Eq. 1) – giving it a chance to course-correct. However, we note that a couple of works offers a theoretical analysis: a) MESS+ (Woisetschläger et al., 2025), incorporates a feedback mechanism that counts the historical constraint violations, and incorporates that into emphasizing/de-emphasizing the corresponding constraint in future decisions – which allows them to bound the number of constraint violations (see Theorem 1); b) PORT (Wu & Silwal), also disclosed in late-October '25, makes certain assumptions about the efficacy of kNN-based performance and cost predictors (see Assumption 1) in order to guarantee competitive ration of online serving (see Theorem 1). We believe the theoretical analyses assume restricted settings which the present work doesn't consider. However, we leave a thorough and careful analysis of optimality gap/competitive ratio/constraint violation for future work.

A.2 ORACLE PERFORMANCE

Table 11 compares POLLINATOR against the closest baseline, MIRT-Router, as well as an Oracle result where we choose the best and cheapest LLM for each sample .

Table 11: Comparison with Oracle results.

| Method | Datasets | | | | | | | |
|---|---|---|---|---|---|---|---|---|
| | In-Domain | | Out-of-Domain | | MMLU-Pro | | BFCL-V3 (ToolCall) | |
| | Perf (%)↑ | Cost ($)↓ | Perf (%)↑ | Cost ($)↓ | Perf (%)↑ | Cost ($)↓ | Perf (%)↑ | Cost ($)↓ |
| Oracle | 95.52 | 0.19 | 98.32 | 0.03 | 98.17 | 0.79 | 100.0 | 0.01 |
| MIRT-Router | 80.67 | 0.42 | 87.12 | 0.14 | 78.84 | 1.18 | 90.66 | 0.008 |
| POLLINATOR | 81.38 | 0.39 | 87.37 | 0.14 | 79.18 | 0.88 | 92.00 | 0.006 |

## A.3 DATASETS DETAILS

The datasets (Table 12) span a wide range of domains and task categories. In-domain datasets include reasoning, code, and QA tasks. Out-of-domain datasets test generalization to unseen tasks. Additional benchmarks, MMLU-Pro and BFCL V3 (Simple), evaluate more challenging reasoning problems and tool use. Each dataset lists the task type, evaluation metric, and train/test sizes.

Table 12: Details of in-domain, out-of-domain, and additional datasets used in our experiments.

| **In-domain** | | | | |
|---|---|---|---|---|
| **Dataset** | **Type** | **Evaluation Metric** | **Train Size** | **Test Size** |
| ACLUE (Zhang & Li, 2023) | Ancient Chinese | accuracy | 1400 | 600 |
| ARC_C (Clark et al., 2018) | Reasoning | accuracy | 1400 | 600 |
| CMMLU (Li et al., 2024) | Chinese Multitask | accuracy | 7000 | 3000 |
| Hotpot_QA (Yang et al., 2018) | Multi-Hop | EM | 1400 | 600 |
| MATH (Hendrycks et al., 2021) | Math | accuracy | 1400 | 600 |
| MBPP (Austin et al., 2021) | Code | pass@1 | 630 | 270 |
| MMLU (Hendrycks et al., 2020) | Multitask | accuracy | 9800 | 4200 |
| SQuAD (Rajpurkar et al., 2018) | Reading Comprehension | f1 | 1400 | 600 |
| **Out-of-domain** | | | | |
| **Dataset** | **Task type** | **Evaluation Metric** | **Train Size** | **Test Size** |
| CEVAL (Huang et al., 2024) | Chinese Multitask | accuracy | - | 1000 |
| CommonsenseQA (Talmor et al., 2019) | Commonsense Reasoning | accuracy | - | 1000 |
| GSM8K (Cobbe et al., 2021) | Math | accuracy | - | 1000 |
| HumanEval (Chen et al., 2021) | Code | pass@1 | - | 160 |
| **Additional Datasets** | | | | |
| MMLU-Pro (Wang et al., 2024) | Multitask Reasoning | accuracy | 9602 | 2430 |
| BFCL V3 (Simple) (Patil et al., 2025) | Tool-Use / Function Calling | accuracy | 125 | 75 |

## A.4 CANDIDATE LLMs FOR VARIOUS DATASETS

For our routing experiments, we select a set of 20 representative LLMs as candidates for in-domain and out-of-domain datasets (see Table 17). The candidate LLMs, along with their input and output costs, for MMLU-Pro and BFCL-V3 (ToolCall) are reported in Tables 18 and 19, respectively.

## A.5 AVERAGE PERFORMANCE–COST CHARACTERISTICS OF INDIVIDUAL LLMs

To understand the standalone efficiency of each model, we report the average performance and total cost of all LLMs across the datasets. Table 13 presents results on the In-Domain dataset. Here, DeepSeek-Chat and DeepSeek-Coder emerge as the strongest models, closely followed by large models such as Qwen2.5-72B-Instruct and GLM-4-Plus. In contrast, smaller or task-specialized models (e.g., Qwen2.5-Math-7B-Instruct) show lower average performance, reflecting their narrow training scope. Table 14 reports the same statistics for the Out-of-Domain dataset. The relative ordering remains broadly consistent: low-cost 7B–8B models offer attractive price points but lag in accuracy compared to larger 32B–72B models, while DeepSeek models again strike a strong accuracy–cost balance. Table 15 summarizes performance and total cost on MMLU-Pro. This shows high-end frontier models such as O4-Mini, GPT-4.1, and Claude-3.5-Sonnet provide superior general reasoning performance but at a substantially higher cost. Finally, Table 16 presents results for BFCL-V3 (ToolCall), which shows Gemini-1.5-Flash, GPT-4.1-Nano, and GPT-4o-Mini models deliver strong accuracy at low cost (BFCL-V3 evaluation containing only a small number of test queries, which keeps total cost minimal).

Table 13: Average performance and cost of individual LLMs on In-Domain data (sorted in ascending order of cost).

| Model | Performance (%) | Cost ($) |
|---|---|---|
| Qwen2.5-7B-Instruct | 61.38 | 0.0607 |
| Llama3.1-8B-Instruct | 31.86 | 0.0877 |
| Mistral-7B-Instruct-v0.2 | 10.42 | 0.1123 |
| Qwen2.5-32B-Int4 | 78.27 | 0.1380 |
| Ministral-8B-Instruct-2410 | 48.70 | 0.3112 |
| Qwen2.5-Math-7B-Instruct | 9.80 | 0.3447 |
| Gemini1.5-Flash | 72.90 | 0.3528 |
| GLM-4-Air | 72.00 | 0.3563 |
| DeepSeek-Coder | 80.61 | 0.4695 |
| DeepSeek-Chat | 80.74 | 0.4740 |
| GPT-4o-Mini | 70.67 | 0.7225 |
| GPT-4o-Mini+CoT | 71.71 | 1.6783 |
| Mixtral-8x7B-Instruct | 34.76 | 1.9891 |
| Llama3.1-70B-Instruct | 71.11 | 2.4712 |
| Qwen2.5-72B-Instruct | 79.97 | 2.4793 |
| QwQ-32B-Preview | 59.73 | 8.3434 |
| Llama3.1-405B-Instruct | 77.54 | 10.2818 |
| GPT-4o | 77.53 | 12.9362 |
| GLM-4-Plus | 79.02 | 19.1334 |

Table 14: Average performance and cost of models on Out-of-Domain data (sorted in ascending order of cost).

| Model | Performance (%) | Cost ($) |
|---|---|---|
| Qwen2.5-7B-Instruct | 59.94 | 0.0147 |
| Llama3.1-8B-Instruct | 44.62 | 0.0256 |
| Mistral-7B-Instruct-v0.2 | 12.15 | 0.0344 |
| Qwen2.5-32B-Int4 | 87.25 | 0.0463 |
| Qwen2.5-Math-7B-Instruct | 32.85 | 0.0805 |
| GLM-4-Air | 73.54 | 0.0940 |
| Gemini1.5-Flash | 71.99 | 0.1013 |
| Ministral-8B-Instruct-2410 | 59.83 | 0.1112 |
| DeepSeek-Coder | 86.33 | 0.1504 |
| DeepSeek-Chat | 86.39 | 0.1511 |
| GPT-4o-Mini | 80.79 | 0.2928 |
| Mixtral-8x7B-Instruct | 27.82 | 0.5038 |
| GPT-4o-Mini+CoT | 80.70 | 0.5109 |
| Llama3.1-70B-Instruct | 75.98 | 0.6830 |
| Qwen2.5-72B-Instruct | 86.08 | 0.7542 |
| QwQ-32B-Preview | 76.30 | 2.3498 |
| Llama3.1-405B-Instruct | 82.06 | 2.9023 |
| GPT-4o | 84.87 | 5.2990 |
| GLM-4-Plus | 87.03 | 5.4369 |

## A.6   PRICING OF CANDIDATE LLMS

We report both input and output token pricing ($/1M tokens) for all candidate models. Candidate LLMs exhibit drastic variation in pricing. Table 17 summarizes the base set of LLMs for in-domain and out-of-domain datasets, while Table 18 and Table 19 details the pricing of models used for MMLU-Pro and BFCL ToolCalling benchmarks.

Table 15: Average performance and cost of models on MMLU-Pro, sorted in ascending order of cost.

| Model | Performance (%) | Cost ($) |
|---|---|---|
| Gemini-1.5-Flash-002 | 63.80 | 0.3316 |
| Gemini-2.0-Flash-Exp | 78.10 | 0.8765 |
| GPT-4.1-Nano | 61.68 | 0.8847 |
| Meta-Llama-3.1-70B | 53.08 | 2.1059 |
| GPT-4.1-Mini | 78.55 | 2.3568 |
| Meta-Llama-3-70B-Instruct | 54.95 | 2.3778 |
| Meta-Llama-3.1-70B-Instruct | 63.67 | 3.5321 |
| Meta-Llama-3-70B | 53.37 | 3.7496 |
| Claude-3.5-Haiku (2024-10-22) | 61.47 | 4.5741 |
| Gemini-1.5-Pro-002 | 71.61 | 5.0187 |
| O4-Mini | 81.67 | 5.5064 |
| GPT-4.1 | 79.47 | 11.7164 |
| Claude-3.5-Sonnet (2024-10-22) | 77.81 | 17.4792 |
| Claude-3.5-Sonnet | 77.35 | 18.4651 |

Table 16: Average performance and cost of LLMs on BFCL-V3 (Toolcall) (sorted in ascending order of cost).

| Model | Performance (%) | Cost ($) |
|---|---|---|
| Gemini-1.5-Flash | 88.00 | 0.0049 |
| GPT-4.1-Nano | 77.33 | 0.0059 |
| GPT-4o-Mini | 90.67 | 0.0088 |
| Gemini-2.0-Flash | 93.33 | 0.0093 |
| GPT-4.1-Mini | 84.00 | 0.0236 |
| Gemini-1.5-Pro | 93.33 | 0.0773 |
| O4-Mini | 86.67 | 0.1100 |
| GPT-4.1 | 94.67 | 0.1177 |
| GPT-4o | 96.00 | 0.1471 |
| O1 | 89.33 | 1.8584 |

Table 19: Pricing details of candidate LLMs selected for BFCL Toolcalling ($/1M tokens).

| LLM | Input $/1M | Output $/1M |
|---|---|---|
| GPT-4o | 2.50 | 10.0 |
| GPT-4o-Mini | 0.15 | 0.60 |
| o1 | 15.0 | 60.0 |
| GPT-4.1-Nano | 0.10 | 0.40 |
| Gemini-1.5-Flash | 0.08 | 0.30 |
| Gemini-1.5-Pro | 1.25 | 5.00 |
| Gemini-2.0-Flash | 0.15 | 0.60 |
| GPT-4.1 | 2.00 | 8.00 |
| GPT-4.1-Mini | 0.40 | 1.60 |
| o4-Mini | 1.10 | 4.40 |

## A.7 ADDITIONAL INTERPRETABILITY RESULTS

**Alignment Between Discrimination Vectors and LLM Abilities.** Table 20 demonstrates that the discrimination vectors ($\alpha_i$) learned by POLLINATOR align strongly with the ability profiles ($\theta_j$) of individual LLMs. Comparing Qwen2.5-7B-Instruct with its math-specialized variant, Qwen2.5-Math-7B-Instruct, we find that queries, with higher mean routing probability, are directed to the general model on in-domain tasks overall, while math-focused queries (e.g., from MATH and GSM8K) are preferentially routed to the specialized model. Conversely, non-math queries from datasets such

Table 17: Pricing details of different LLMs ($/1M tokens) selected for In-Domain and Out-of-Domain Datasets.

| LLM | Input $/1M | Output $/1M |
|---|---|---|
| DeepSeek-Chat | 0.14 | 0.28 |
| DeepSeek-Coder | 0.14 | 0.28 |
| Gemini-1.5-Flash | 0.075 | 0.30 |
| GLM-4-Air | 0.137 | 0.137 |
| GLM-4-Flash | 0.0137 | 0.0137 |
| GLM-4-Plus | 6.85 | 6.85 |
| GPT-4o | 2.50 | 10.0 |
| GPT-4o-Mini | 0.15 | 0.60 |
| GPT-4o-Mini+CoT | 0.15 | 0.60 |
| Llama3.1-8B-Instruct | 0.10 | 0.20 |
| Llama3.1-70B-Instruct | 0.792 | 0.792 |
| Llama3.1-405B-Instruct | 3.15 | 3.15 |
| Ministral-8B-Instruct-2410 | 0.10 | 0.20 |
| Mistral-7B-Instruct-v0.2 | 0.10 | 0.20 |
| Mixtral-8x7B-Instruct | 0.54 | 0.54 |
| Qwen2.5-32B-Instruct-GPTQ-Int4 | 0.10 | 0.20 |
| Qwen2.5-7B-Instruct | 0.10 | 0.20 |
| Qwen2.5-72B-Instruct | 1.08 | 1.08 |
| Qwen2.5-Math-7B-Instruct | 0.10 | 0.20 |
| QwQ-32B-Preview | 1.20 | 1.20 |

Table 18: Pricing details of candidate LLMs selected for MMLU-Pro ($/1M tokens).

| LLM | Input $/1M | Output $/1M |
|---|---|---|
| Claude-3.5-Sonnet | 3.00 | 15.00 |
| Gemini-1.5-Pro | 1.25 | 5.00 |
| Llama3.1-70B | 0.60 | 0.60 |
| Llama3.1-70B-Instruct | 1.00 | 1.00 |
| Llama3-70B | 0.65 | 2.75 |
| Llama3-70B-Instruct | 0.59 | 0.79 |
| Claude-3.5-Sonnet-(alt) | 3.00 | 15.00 |
| Claude-3.5-Sonnet-2024 | 3.00 | 15.00 |
| Claude-3.5-Haiku-2024 | 0.80 | 4.00 |
| Gemini-1.5-Flash | 0.08 | 0.30 |
| Gemini-2.0-Flash | 0.15 | 0.60 |
| GPT-4.1-Nano | 0.10 | 0.40 |
| GPT-4.1 | 2.00 | 8.00 |
| GPT-4.1-Mini | 0.40 | 1.60 |
| o4-Mini | 1.10 | 4.40 |

as MMLU and CommonsenseQA, are mostly routed to the general model. This indicates that the learned discrimination vectors ($\alpha_i$) capture model-specific strengths and effectively guide query allocation.

**LLM Ability and Query Difficulty.** Figure 5 highlights consistent performance improvements of Llama3.1-405B-Instruct over its smaller counterpart. Figure 6 illustrates example queries with predicted $\beta_i$, highlighting strong agreement with human labels. These qualitative cases further demonstrate that the learned routing signals capture meaningful task difficulty. Overall, the results underscore the reliability of our scoring mechanism across diverse query types.

Table 20: Mean predicted routing probability of the general (Qwen2.5-7B-Instruct) vs. math-specialized (Qwen2.5-Math-7B-Instruct) models. Math queries are routed more often to the specialized model, while non-math queries favor the general model.

| Task | Qwen2.5-7B-Instruct | Qwen2.5-Math-7B-Instruct |
|---|---|---|
| All ID Tasks | **0.60** | 0.14 |
| MATH | 0.32 | **0.70** |
| GSM8K | 0.37 | **0.44** |
| MMLU | **0.60** | 0.04 |
| CommonsenseQA | **0.74** | 0.07 |

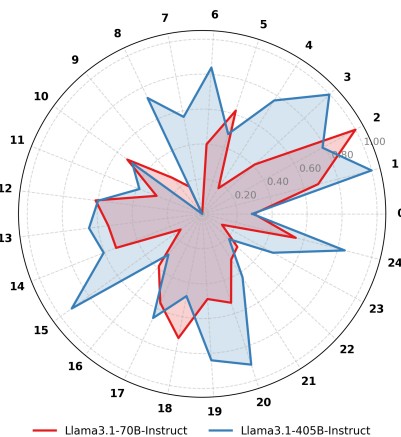

Figure 5: Estimated ability profiles ($\theta_j$) over 25 dimensions. Comparison of Llama3.1-70B-Instruct and Llama3.1-405B-Instruct, showing consistent gains for the larger model.

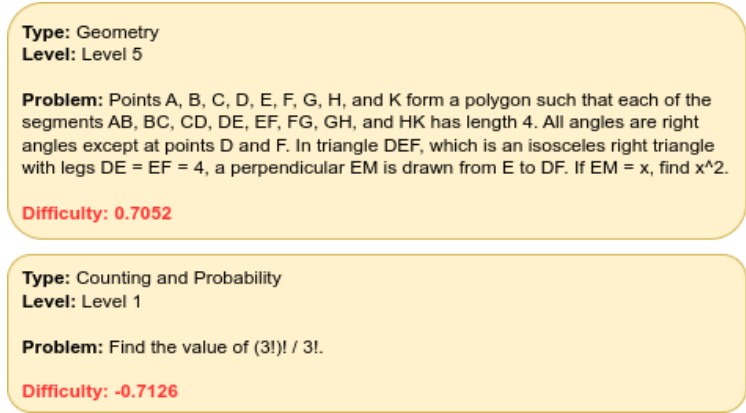

Figure 6: Example queries with predicted $b_i$ values compared to human labels, illustrating the close correspondence between predicted and true difficulty.

## A.8 RUNTIME AND COMPLEXITY ANALYSIS

The complexity of Algorithm 1 is dominated by the sorting operation in Line 4. For a model pool of size $N$, naive sorting takes $\mathcal{O}(N \log N)$ time. The remaining operations are linear, $\mathcal{O}(N)$, as follows: **Predictor Invocation (Line 3):** Fetching ex-ante performance and cost predictions requires $\mathcal{O}(N)$ time.

**Utility Computation (Line 4):** Calculating the utilities $\{u_{ij}\}_{j=1}^{N}$ also takes $\mathcal{O}(N)$ time (see Sec. 2.2).

**Utility Sorting (Line 4):** Sorting the computed utilities is the most expensive step, with complexity

$\mathcal{O}(N \log N)$, dominating the overall runtime.

**Iterative Thresholding (Lines 6–12):** In the worst case, the loop scans all utilities, taking $\mathcal{O}(N)$ time.

**Primal Serving Scheme (Line 14):** Constructing the primal serving scheme requires an additional $\mathcal{O}(N)$ pass.

### A.9 DETAILED ABLATION STUDY OF THE PREDICTOR

Table 21 presents a detailed ablation of our predictor (without the optimizer), analyzing the impact of ability dimension ($\theta$), number of neighbors ($k$), edge-weighting, masking ratio, and embedder selection. Performance generally improves with increasing $\theta$ up to an optimal point. Across different configurations, our predictor consistently outperforms baseline methods, demonstrating robustness to design choices.

### A.10 METHODOLOGY

The overall training and inference flows of POLLINATORare illustrated in Figure 7, showing the dual-tower encoder and IRT-based prediction head used to generate serving plans.

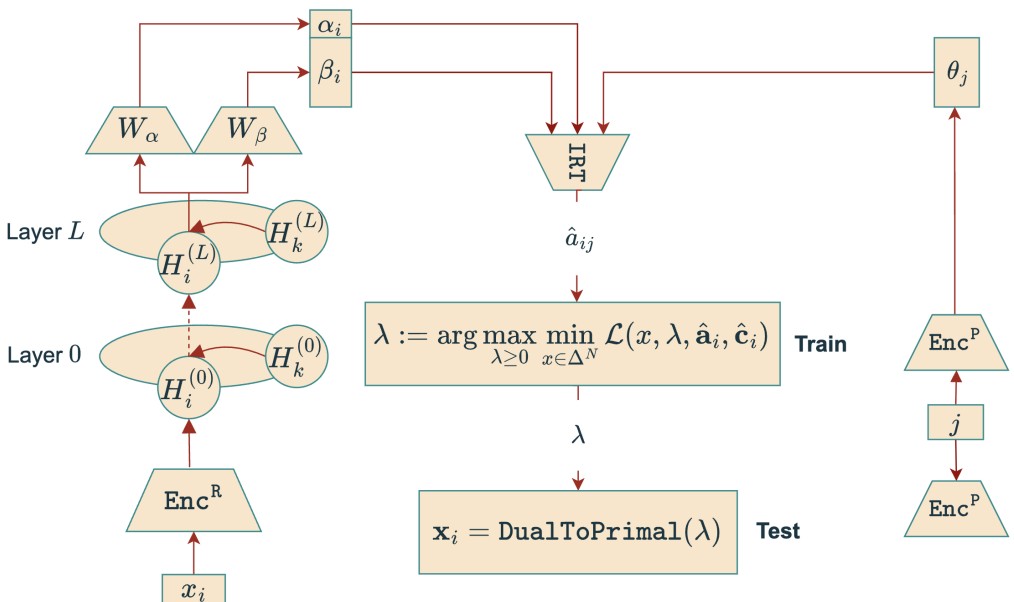

Figure 7: POLLINATOR: Train & Inference Flows. The left tower encodes request $i$ with a GCN with $L$ layers. The right tower encodes producer $j$. The bespoke IRT-based head combines the outputs of the two towers to generate ex ante predictions. In train flow, $\lambda$ is the result of optimization on a held-out validation set, which, during inference, is used to compute the primal serving plan $\mathbf{x}_i$ via Algorithm 1.

### A.11 CANDIDATE LLMS PROFILE DESCRIPTIONS

Table 22 lists the candidate large LLMs used in our experiments for MMLU-Pro & BFCL-V3, along with their key profile descriptions.

## B ADDITIONAL RELATED WORKS

**Item Response Theory.** Item Response Theory (IRT) (Woodruff & Hanson, 1996) models the interaction between latent human ability and item difficulty via logistic functions, ensuring interpretability through monotonicity. Extensions such as MIRT (Reckase, 2006) and neural variants (e.g., NCDM (Wang et al., 2020)) capture richer interactions. Beyond education, IRT has been applied to model evaluation (Liu et al., 2024), recommendation (Liu et al., 2023), leaderboard ranking

Table 21: Ablation study of the predictor (without optimizer). Columns indicate ability ($\theta_j$), dimension ($D$), number of neighbors ($k$), edge-weight, node label masking, request encoder ($\text{Enc}^{\text{R}}$), and performance. Bold values indicate the best configuration within each block.

| Model | $D$ | $k$ | Edge-weight | Masking Ratio | $\text{Enc}^{\text{R}}$ | Performance |
|---|---|---|---|---|---|---|
| **Baselines** | | | | | | |
| Small LLM | – | – | ✗ | ✗ | – | 48.70% |
| Large LLM | – | – | ✗ | ✗ | – | 77.53% |
| KNN-Router | – | 5 | ✗ | ✗ | – | 74.38% |
| MIRT-Router | – | – | ✗ | ✗ | – | 81.17% |
| NIRT-Router | – | – | ✗ | ✗ | – | 75.26% |
| **POLLINATOR: Ablation on $D$ and $k$ (no edge-weight)** | | | | | | |
| | 3 | 3 | ✗ | ✗ | bert-base-uncased | 71.04% |
| | 5 | 3 | ✗ | ✗ | bert-base-uncased | 80.51% |
| | 8 | 3 | ✗ | ✗ | bert-base-uncased | 79.26% |
| | 16 | 3 | ✗ | ✗ | bert-base-uncased | 82.08% |
| | 16 | 5 | ✗ | ✗ | bert-base-uncased | 81.73% |
| POLLINATOR | 16 | 10 | ✗ | ✗ | bert-base-uncased | 81.94% |
| | 16 | 20 | ✗ | ✗ | bert-base-uncased | 82.03% |
| | 25 | 3 | ✗ | ✗ | bert-base-uncased | 82.28% |
| | 25 | 5 | ✗ | ✗ | bert-base-uncased | 82.24% |
| | 25 | 10 | ✗ | ✗ | bert-base-uncased | 82.13% |
| | 25 | 20 | ✗ | ✗ | bert-base-uncased | 81.53% |
| **POLLINATOR: Edge-weight, varying $D$ and $k$** | | | | | | |
| | 3 | 3 | ✓ | ✗ | bert-base-uncased | 71.43% |
| | 5 | 3 | ✓ | ✗ | bert-base-uncased | 80.49% |
| | 8 | 3 | ✓ | ✗ | bert-base-uncased | 79.26% |
| | 16 | 3 | ✓ | ✗ | bert-base-uncased | 82.07% |
| | 25 | 3 | ✓ | ✗ | bert-base-uncased | 81.96% |
| | 35 | 3 | ✓ | ✗ | bert-base-uncased | 80.52% |
| | 45 | 3 | ✓ | ✗ | bert-base-uncased | 81.25% |
| POLLINATOR | 3 | 5 | ✓ | ✗ | bert-base-uncased | 71.12% |
| | 5 | 5 | ✓ | ✗ | bert-base-uncased | 80.27% |
| | 8 | 5 | ✓ | ✗ | bert-base-uncased | 79.37% |
| | 16 | 5 | ✓ | ✗ | bert-base-uncased | 81.90% |
| | 25 | 5 | ✓ | ✗ | bert-base-uncased | 81.20% |
| | 35 | 5 | ✓ | ✗ | bert-base-uncased | 80.53% |
| | 45 | 5 | ✓ | ✗ | bert-base-uncased | 80.72% |
| **POLLINATOR: Node Masking** | | | | | | |
| | 16 | 3 | ✓ | 0.1 | bert-base-uncased | 81.96% |
| | 16 | 3 | ✓ | 0.2 | bert-base-uncased | 82.01% |
| | 16 | 3 | ✓ | 0.3 | bert-base-uncased | 81.97% |
| POLLINATOR | 16 | 3 | ✓ | 0.5 | bert-base-uncased | 81.82% |
| | 25 | 5 | ✓ | 0.1 | bert-base-uncased | 81.79% |
| | 25 | 5 | ✓ | 0.2 | bert-base-uncased | 81.89% |
| | 25 | 5 | ✓ | 0.3 | bert-base-uncased | 82.12% |
| | 25 | 5 | ✓ | 0.5 | bert-base-uncased | 81.68% |
| **POLLINATOR: Different $\text{Enc}^{\text{R}}$** | | | | | | |
| | 16 | 3 | ✗ | ✗ | all-MiniLM-L6-v2 | 80.86% |
| | 25 | 3 | ✗ | ✗ | all-MiniLM-L6-v2 | 80.74% |
| | 16 | 3 | ✗ | ✗ | all-mpnet-base-v2 | 81.70% |
| | 25 | 3 | ✗ | ✗ | all-mpnet-base-v2 | 82.37% |
| | 25 | 3 | ✗ | ✗ | text-embedding-3-large | 81.97% |
| POLLINATOR | 16 | 3 | ✓ | ✗ | all-mpnet-base-v2 | 81.70% |
| | 16 | 3 | ✓ | ✗ | all-MiniLM-L6-v2 | 80.75% |
| | 16 | 3 | ✓ | ✗ | text-embedding-3-large | 80.15% |
| | 25 | 5 | ✓ | ✗ | all-mpnet-base-v2 | 82.60% |
| | 25 | 5 | ✓ | ✗ | all-MiniLM-L6-v2 | 80.74% |
| | 25 | 5 | ✓ | ✗ | text-embedding-3-large | 81.91% |

(Rodriguez et al., 2021), and LLM assessment (Guinet et al., 2024; Liu et al., 2025). We adopt IRT for its interpretability and proven effectiveness in human and machine assessment.

**LLM Routers.** LLM routing seeks to assign queries to the most suitable model for optimal accuracy–cost tradeoffs. Early works like FrugalGPT (Chen et al., 2023) and AutoMix (Aggarwal et al.,

Table 22: List of candidate LLMs with their profile descriptions.

| LLM Name | Profile Description |
|---|---|
| Gemini 2.0 Flash | Released on Dec 11, 2024 by Google DeepMind. Experimental version of Gemini 2.0 Flash, focusing on enhanced speed and performance. Features include a Multimodal Live API for real-time audio and video interactions, improved spatial understanding, native image and controllable text-to-speech with watermarking, and integrated tool use, including Google Search. Also introduces improved agentic capabilities and a new Google Gen AI SDK. |
| Gemini 1.5 Pro 002 | Released on Sep 24, 2024 by Google DeepMind. Updated Gemini 1.5 Pro with a 2M token context window and up to 8,192 token outputs. Designed for diverse tasks via Google AI Studio and Vertex AI. |
| Meta-Llama-3.1-70B | Released on Jul 23, 2024 by Meta AI. A 70B parameter model pre-trained on 15T tokens from public sources, designed for advanced language understanding, coding, and reasoning. |
| Claude 3.5 Haiku | Released on Oct 22, 2024 by Anthropic. Optimized for efficiency and speed with a 200K token context window and 8,192 token outputs. Suitable for rapid-response tasks. |
| Meta-Llama-3.1-70B-Instruct | Released on Jul 23, 2024 by Meta AI. Instruction-tuned variant of Llama 3.1-70B, fine-tuned on public datasets and 10M+ human annotations to enhance instruction-following. |
| GPT-4.1 Nano | Released on Apr 14, 2025 by OpenAI. Compact GPT-4.1 version for on-device tasks with reduced compute needs while maintaining strong performance. |
| Gemini 1.5 Flash 002 | Released on Sep 24, 2024 by Google DeepMind. Updated Gemini 1.5 Flash with a 1M token context window and up to 8,192 token outputs. Optimized for speed and cost-efficiency. |
| o4-Mini | Released on Aug 6, 2024 by OpenAI. A smaller GPT-4o variant with 1,047,476 token context window and 32,768 token outputs, offering efficiency while retaining robust performance. |
| Claude 3.5 Sonnet | Released on Oct 22, 2024 by Anthropic. Balances performance and efficiency with a 200K token context window and 8,192 token outputs. General-purpose model. |
| Claude 3.5 Sonnet | Released on Oct 22, 2024 by Anthropic. Part of the Claude 3.5 series, offering 200K context window, optimized for diverse tasks. Achieved 59.1% on GPQA Diamond benchmark. |
| Meta-Llama-3-70B | Released on Apr 18, 2024 by Meta AI. A 70B parameter model pre-trained on 15T tokens, designed for high-performance language tasks. |
| Meta-Llama-3-70B-Instruct | Released on Apr 18, 2024 by Meta AI. Instruction-tuned version of Llama 3-70B, aligned with user queries via public datasets and 10M+ annotations. |
| GPT-4.1 | Released on Apr 14, 2025 by OpenAI. Enhanced GPT-4 variant with improved reasoning, coding, and agentic abilities. |
| GPT-4.1 Mini | Released on Apr 14, 2025 by OpenAI. Smaller GPT-4.1 variant, optimized for efficiency while maintaining strong task performance. |
| GPT-4.1-Nano | Released on April 14, 2025, by OpenAI. A compact version of the GPT-4.1 model, designed for on-device tasks with reduced computational requirements. Maintains strong performance across various benchmarks while being optimized for efficiency. |
| GPT-4o | GPT-4o: Released on November 20, 2023, by OpenAI. A large language model capable of handling complex tasks requiring deep understanding of language. Features include advanced reasoning capabilities, multimodal capabilities, and a context window of up to 128,000 tokens. Available through OpenAI's API. |
| GPT-4o-Mini | GPT-4o Mini: Released on November 20, 2023, by OpenAI. A smaller variant of the GPT-4o model, designed for efficiency while maintaining strong performance across various tasks. Optimized for applications requiring reduced computational resources. |
| o1 | Released on August 16, 2024, by OpenAI. A large language model capable of handling complex tasks requiring deep understanding of language. Features include advanced reasoning capabilities, multimodal capabilities, and a context window of up to 128,000 tokens. Available through OpenAI's API. |

2023) use cascaded inference, while later methods train lightweight routers such as HybridLLM (Ding et al., 2024), RouteLLM (Ong et al., 2024), and Zooter (Lu et al., 2023). RouterDC (Chen et al., 2024) and KNN-based approaches (Hu et al., 2024) further reduce costs, and GraphRouter (Feng et al., 2024) leverages GNNs but depends on task priors. EmbedLLM (Zhuang et al., 2025) learns compact embeddings via matrix factorization to support routing at scale. Commercial systems like Martian[15] and Neutrino AI[16] demonstrate practical benefits, reporting major savings. Unlike these, our approach couples difficulty-aware estimation with online dual optimization, yielding interpretable and cost-efficient routing.

---

[15]https://withmartian.com
[16]https://neutrinoapp.com

**Graph-based Modeling.** Graphs naturally capture relational structures (Fey et al., 2023; Cao et al., 2023; Chen et al., 2022). Classical methods like label propagation (Xie et al., 2022) leverage edges for transductive learning, while GNNs (Kipf, 2016; Hamilton et al., 2017) extend message passing to learn expressive representations. Recent work highlights their zero-/few-shot potential (Fey et al., 2023; Cao et al., 2023) in domains such as recommendation and social networks. Building on these advances, we employ GNNs to design the predictor of POLLINATOR [17]

---

[17]All code and datasets for POLLINATOR are provided in the supplementary material.

