# OpenReview forum: "POLLINATOR: OPTIMAL MATCHMAKING IN AN INTELLIGENCE MARKETPLACE"
_ICLR.cc/2026/Conference — Submitted to ICLR 2026_

### Official Review · Reviewer_toZC · 2025-10-22

**Soundness:** 3
**Presentation:** 3
**Contribution:** 3
**Rating:** 6
**Confidence:** 2

**Summary:**

This paper proposes an elastic model selection framework that dynamically allocates large models to optimize performance and cost. The idea is interesting and practically valuable for large-scale deployment. However, the paper lacks clarity on how dynamic model switching is feasible in real settings, given that invoking multiple cloud or local services incurs significant latency and token overhead. Moreover, the method’s handling of prediction–optimization mismatch is insufficiently analyzed, and scalability issues related to operational and inference costs are not well quantified. Overall, the paper presents a promising direction but requires deeper technical explanation and broader empirical validation to be fully convincing.

**Strengths:**

The paper addresses an important and timely problem in efficient large-model deployment by proposing an elastic selection mechanism to balance accuracy and cost. Its idea of dynamically routing tasks to different models based on predicted utility is conceptually appealing and reflects practical system needs. The formulation is well-motivated, and the integration of optimization principles into model selection demonstrates good technical insight.

**Weaknesses:**

The main limitation of the paper lies in the practicality and scalability of its proposed elastic model routing mechanism. Although the idea of adaptively selecting large models is appealing, the paper lacks a concrete analysis of how such routing can be implemented efficiently in real deployment scenarios. Frequent switching between models or services would inevitably introduce latency, overhead, and token costs, yet these factors are not sufficiently quantified or discussed.
﻿
Moreover, the framework’s predictive component relies heavily on estimated performance and cost metrics without a clear strategy for handling uncertainty or prediction errors. The absence of robustness tests, sensitivity analysis, and ablation studies weakens the empirical evidence. Additionally, operational constraints such as service rate limits, model drift, or dynamic pricing are not modeled or evaluated, raising concerns about the system’s stability and reliability under real-world conditions.

**Questions:**

1. How does the proposed elastic model selection framework manage the latency and token overhead introduced by dynamically invoking multiple large models across cloud or local services?
﻿
2. What mechanisms are used to mitigate prediction–optimization mismatch in the estimated performance and cost metrics ? Are calibration or uncertainty-aware strategies incorporated during training or inference?
﻿
3. Can the authors provide additional experiments or analysis to evaluate scalability, particularly in terms of end-to-end latency, system throughput, and robustness under dynamic model pricing or service availability changes?

---

> ### Author Response · Authors · 2025-11-22
> **Point-by-point responses to Reviewer toZC**
>
> Thank you for recognizing the motivation, practical relevance, and technical soundness of our framework. We address your specific comments below.
>
> - **How does the proposed elastic model selection framework manage the latency and token overhead introduced by dynamically invoking multiple large models across cloud or local services?**
>
> **Response:**
> It’s worth noting that the prompt router routes each request to exactly one LLM, either hosted by a provider (e.g., OpenAI, TogetherAI) or self-hosted (e.g., with vLLM) – and in no cases more than one LLM is being invoked.
>
> As noted in the response to a question below, the prompt router incurs sub-100ms p99 latency, which is negligible, given that self-hosted LLMs take ~400-700ms (depends on the parameter count, architecture and the inference engine – and varies across workloads), and those hosted by provider often exceeds ~1.2s (even with provisioned throughput, such as PTU in Azure).
>
> There are no additional token overhead, as such, beyond those already accounted for under the prompt router latency.
>
> - **What mechanisms are used to mitigate prediction–optimization mismatch in the estimated performance and cost metrics ? Are calibration or uncertainty-aware strategies incorporated during training or inference?**
>
> **Response:**
> Calibration is integrated into inference: the calibrated probabilities are fed to the primal problem (Eq. 1), as well as the dual serving plan (Algorithm 1).
> There are no special treatments for epistemic and aleatoric uncertainty in the current work. However, it is worth noting that the superior performance in Out-of-Domain dataset (see **Table 2**) indicates its robustness to uncertainty (at least those caused by outliers/OOD data). We defer a careful treatment of uncertainty to future work.
>
> - **Can the authors provide additional experiments or analysis to evaluate scalability, particularly in terms of end-to-end latency, system throughput, and robustness under dynamic model pricing or service availability changes?**
>
> **Response:**
> We thank the reviewer for asking these pertinent questions. Please see our response below:-
>
> **End-to-End Latency.** The bottleneck is the nearest neighbour look-up. Even an exact nearest neighbour search yields sub-100ms p99 (99th percentile) latency, as seen in the table below (also added as **Table 7** in the **Appendix A.1**). Note that with approximate nearest neighbour search, especially with HNSW index, the sub-100ms p99 latency can be maintained in industry-scale data (a study [1] from Pinecone reports 50ms p99 at a good recall). It is worth mentioning that the largest dataset reported in the literature on prompt routing has cardinality ~1.5M (source [2]), which is considered “small” in the parlance of approximate nearest neighbour search.
>
>
>
> | Dataset            | Per-Query Inference Time (ms) |
> | :----------------- | ---------------------:|
> | In-Domain | 69.07|
> | Out-of-Domain | 78.59 |
> | MMLU-Pro  | 20.37  |
> | BFCL-V3 (ToolCall) | 2.66 |
>
>
>
> **Throughput.** Industrial vector databases, such as Qdrant, can support 1,200 RPS (Requests Per Second) at ~0.99 precision (source [3]). Thus, instead of nearest neighbour lookup, the throughput bottleneck shifts to the web-framework. FastAPI, with its asynchronous async/await primitives, offers high throughput suitable for an industry-grade prompt router – which, moreover, is horizontally scalable. Note that in the current work, we limit our scope to the functional requirements of prompt router, not its non-functional requirements, such as latency/throughput.
>
> **Robustness w.r.t. Dynamic Pricing.** The rate cards for providers change infrequently. However, constructs such as provisioned throughput (e.g., PTU [4] in Azure Foundry) render the rate card a function of throughput. Even in this case, organizations typically purchase a fixed amount of PTU, rendering the rate card essentially frozen over the contract period (an year). In the (infrequent) event of change in rate card, the dual variables need to be recomputed and deployed via a configuration service to all workers executing Algorithm 1.
>
> **Robustness w.r.t. Availability.** The providers indeed suffer downtimes, and to counteract that, one typically routes to a fallback (which is typically the next available provider in the ordered list, $x_{ij}$ – Line 2 in Algorithm 1) after a pre-configured amount of retry. The provision of fallback has been popularized by commercial prompt routers, such as OpenRouter (API reference [5]).
>
> We have included the above latency and robustness analysis in **Appendix A.1** in the revised manuscript.
>
> [1] https://www.pinecone.io/learn/series/faiss/hnsw/
>
> [2] Feng. et al. "IPR: Intelligent Prompt Routing with User-Controlled Quality-Cost Trade-offs
>
> [3] https://qdrant.tech/benchmarks/
>
> [4]https://learn.microsoft.com/en-us/azure/ai-foundry/openai/concepts/provisioned-throughput?view=foundry-classic&tabs=global-ptum
>
> [5]https://openrouter.ai/docs/features/provider-routing

---

> > ### Author Response · Authors · 2025-11-25
> > **Looking forward to your feedback**
> >
> > Dear Reviewer,
> >
> > Thank you for your valuable feedback and constructive comments.
> >
> > In our rebuttal, we have included additional experiments and enhanced explanations, and we hope that we have addressed all the concerns you raised. We remain open to further discussion and would be happy to clarify any remaining questions.
> >
> > This is just a gentle reminder. We look forward to hearing from you at your convenience.
> >
> > Thank you.
> >
> > The Authors

---

> ### Author Response · Authors · 2025-12-03
> **Response to further comment of Reviewer toZC**
>
> Please find the response to additional stress-test experiment as suggested:
>
> **I believe that stress tests under different levels of concurrency would more directly reflect the system's performance.**
>
> **Response:** We conducted a realistic load-test with an industry-standard tool, Locust [1], under varying levels of concurrency (at ~57 requests-per-second) to evaluate POLLINATOR’s inference performance under real-world conditions. The inference engine consists of a) FastAPI [2]-based web-framework, b) Infinity [3]-based embedder, and, c) Qdrant [4]-based approximate k-nearest-neighbour search. All calls to embedder and nearest neighbour searches are non-blocking, via Python’s asynchronous coroutine (`async` and `await`) – which mimics the architecture of a typical real-world inference engine.
>
> In the revised manuscript, **Figure 4**  (**Appendix A.1**) presents the overall system dynamics during this load ramp, including achieved throughput, median (p50) and tail (p95) latencies, and active user concurrency.  We additionally report detailed request and response time statistics (refer **Tables 8, 9** in **Appendix A.1**) in Tables below.  The results show that the system sustains stable throughput (**~57 RPS**) with **zero failures**, while maintaining **low median latency** (≈23–28ms) and tightly bounded tail latency (p95 ≤ 55ms) even under peak concurrency. These observations confirm POLLINATOR’s robust scalability and reliable performance under **heavy concurrent workloads**.
>
> ## Response Time Statistics
>
> | Method | Name   | 50%ile (ms) | 60%ile (ms) | 70%ile (ms) | 80%ile (ms) | 90%ile (ms) | 95%ile (ms) | 99%ile (ms) | 100%ile (ms) |
> |--------|--------|---------------|---------------|---------------|---------------|---------------|----------------|----------------|----------------|
> | GET    | /route | 23            | 25            | 28            | 34            | 43            | 55             | 96             | 200            |
> | —      | Aggregated | 23        | 25            | 28            | 34            | 43            | 55             | 96             | 200            |
>
>
> ## Request Statistics
>
> | Type | Name   | # Requests | # Fails | Average (ms) | Min (ms) | Max (ms) | Average size (bytes) | RPS  | Failures/s |
> |------|--------|--------------|-----------|----------------|------------|------------|--------------------------|--------|---------------|
> | GET  | /route | 10,213       | 0         | 27.67          | 12         | 200        | 1182.69                  | 56.73 | 0             |
> | —    | Aggregated | 10,213   | 0         | 27.67          | 12         | 200        | 1182.69                  | 56.73 | 0             |
>
>  [1] https://docs.locust.io/en/stable/index.html
>
>  [2] https://fastapi.tiangolo.com/
>
>  [3] https://docs.langchain.com/oss/python/integrations/text_embedding/infinity
>
>  [4] https://qdrant.tech/

---

### Official Review · Reviewer_ex1Y · 2025-10-31

**Soundness:** 2
**Presentation:** 2
**Contribution:** 2
**Rating:** 4
**Confidence:** 4

**Summary:**

This paper addresses the challenge of optimal LLM selection in the intelligence marketplace in which numerous LLMs with varying cost-performance tradeoffs make it difficult for users to  select the right model according to the request.  The proposed method is a novel router integrating a predictor and an online optimizer, aiming to balance inference performance, cost savings, and real-time serving capability for different scenarios.

**Strengths:**

The paper focuses a critical challenge in the intelligence marketplace. This is a timely problem of practical importance.

It combines GCN) with IRT for prediction, as well as convex optimization for online serving. This integration results into significant performance improvement over existing methods.

Authors have conducted comprehensive experiments that cover a total of 14 datasets. Results demonstrate clear advantages.

**Weaknesses:**

Somehow the modeling comes with oversimplified cost and constraints. The paper only models cost based on token counts, did not take into account other practical factors such as hardware deployment overhead or regional pricing differences, which plays an important role in real-world applications. Other common industrial constraints are not considered in the proposed approach e.g., minimum LLM usage volume commitments.

The paper emphasizes on engineering implementation, it lacks in-depth theoretical analysis and comparisons with RL-based dynamic routing methods.  Also, it assumes static LLM abilities, with no mechanism to adapt to LLM updates e.g., iterative model fine-tuning or sudden shifts in query types.

**Questions:**

Is the proposed method sensitive to the hyper parameter selection ?
What are some alternative methods for cost optimization in LLM deployment?

---

> ### Author Response · Authors · 2025-11-22
> **Point-by-point responses to Reviewer ex1Y**
>
> Thank you for carefully reviewing our submission and highlighting the strengths of our paper. Please find below our responses to the specific comments you made:-
>
> - **Somehow the modeling comes with oversimplified cost and constraints... did not take into account other practical factors .. Other common industrial constraints are not considered ..**
>
>
> **Response:**
>
> **Cost Modeling:**
> It is indeed true that token-usage isn’t the only basis of pricing LLM access. For on-premise deployment of LLM atop an inference engine, infrastructure cost depends on cloud provider, instance type, region, and reservation model (e.g., spot vs. reserved), and large organizations often negotiate preferential rates.
> Similarly, for access via API, the same LLM can be accessed via different providers at different price points (expressed as $-per-million-token). Some providers, such as Azure, sell provisioned throughput (PTU) – which further modulates the rate card.
>
> In this work, we abstract out these nuances by modeling access cost purely via a per-million-token rate card, whose value reflects the unit economics of the current deployment (e.g., GPU count and utilization). When the rate card changes, we re-solve the optimization problem (Eq. 1) and redistribute the updated dual variables to all workers running Algorithm 1.
>
> **Constraint Modeling:**
> We acknowledge that minimum-volume commitment  is a practical consideration. To this end, we note that Proposition 1-2 and Algorithm 1 extends seamlessly to any optimization problem with strongly convex objective and linear constraints (similar to Eq. 1) – only requiring a change in the expression for utility, $u_{ij}$ (see Sec. 2.2). Every new constraint added to Eq. 1 need to be added to $u_{ij}$ with an appropriate sign (depending on whether the constraint is covering- or packing-type) and multiplied by its own Lagrangian dual variable. Furthermore, this framework  also supports other linear constraints, such as p99 latency.
>
> We have incorporated these clarifications into **Section 2.2** of the revised manuscript.
>
> - **The paper emphasizes on engineering implementation, it lacks comparisons with RL-based dynamic routing... it assumes static LLM abilities..no mechanism to adapt to LLM updates..or sudden shifts in query types.**
>
> **Response:**
>
> **Comparison with RL-based Method:** We note the reviewer’s comment regarding RL-based dynamic routing methods.  To the best of our knowledge, the first RL-driven routing approach (Router-R1 [1])  relevant to our setting was published only recently (October 2025). Since it appeared post-submission, we could not include it, but we will consider it in future work.
>
>  **Handling Drift:** It is indeed true that LLM abilities may drift with subsequent releases of frontier models, or may be caused by iterative fine-tuning of self-hosted models. Similarly, since prompts are frequently updated in deployed systems (e.g., addition of new instructions in prompt, addition/deletion of tools, etc.) – drift can occur.
>
> Such drifts are dealt with by re-training the predictors and the (primal) optimizer in POLLINATOR to adapt to the new data distribution. The trigger for re-training (i.e., a drift detection module), however, lies outside POLLINATOR’s system boundary to promote simplicity. Alternatively, one can configure a cron-based periodic re-training.
>
> However, we note that as shown in Table 2, POLLINATOR outperforms the baselines in OOD datasets, thus yielding resilience to data-drift out of the box. We further evaluated data drift on MMLU-Pro (**Appendix A.1, Table 10** in revised PDF).
>
> [1] Zhang et.al "Router-r1: Teaching llms multi-round routing and aggregation via reinforcement learning.
>
> - **Is the proposed method sensitive to the hyper parameter selection?**
>
> **Response:**  Our method is not highly sensitive to hyperparameter choices. As shown in the detailed ablation study provided in **Appendix A.9 (Table 21)**, we systematically vary all key hyperparameters—including ability dimension ($\theta$), number of neighbors ($k$), edge-weighting, masking ratio, and request encoder. Across this broad range of settings, the performance remains stable with only minor fluctuations, and the predictor consistently outperforms all baselines. These results demonstrate that the proposed method is robust to hyperparameter selection.
>
> - **What are some alternative methods for cost optimization in LLM deployment?**
>
> **Response:**  During deployment, additional levers include: a) speculative decoding and b) optimized GPU kernels (e.g., FlashAttention3).
> Post-deployment, cost can be reduced via: a) prompt compression, b) response caching, c) routing to the right-size model, and d) cascades (calling increasingly larger models). While cascades are similar in spirit to routing, they incur higher cost and latency due to multiple sequential model calls. Routing instead estimates correctness without calling any model – offering extremely low cost and latency overhead.

---

> > ### Author Response · Authors · 2025-11-25
> > **Looking forward to your feedback**
> >
> > Dear Reviewer,
> >
> > Thank you for your valuable feedback and constructive comments.
> >
> > In our rebuttal, we have included additional experiments and enhanced explanations, and we hope that we have addressed all the concerns you raised. We remain open to further discussion and would be happy to clarify any remaining questions.
> >
> > This is just a gentle reminder. We look forward to hearing from you at your convenience.
> >
> > Thank you.
> >
> > The Authors

---

### Official Review · Reviewer_NoyV · 2025-10-31

**Soundness:** 3
**Presentation:** 3
**Contribution:** 3
**Rating:** 6
**Confidence:** 3

**Summary:**

This paper presents POLLINATOR, a system for optimal LLM matchmaking in the Intelligence Marketplace. The framework includes two key components: a) Predictor, a data-efficient prediction module that integrates a GCN with an IRT head to improve accuracy while reducing training cost; b) Optimizer, a strongly convex, dual-based optimization module that supports efficient online routing and enforces safety constraints during model selection. Experiments across multiple benchmarks show consistent improvements in cost–performance trade-offs over existing routing methods.

**Strengths:**

- The integration of GCN and IRT for performance prediction is innovative, effectively capturing task–model relationships while offering interpretability via parameters representing model ability and query difficulty.
- The optimization is elegantly formulated as a strongly convex program, enabling closed-form solutions for utility allocation. This avoids costly iterative optimization and supports real-time decision-making with solid theoretical grounding.
- Extensive experiments on 14 diverse benchmarks demonstrate consistent gains in both accuracy and cost efficiency compared with strong baselines.

**Weaknesses:**

- The paper mentions potential mismatch between predicted and actual values but does not quantify its impact on constraint satisfaction or budget adherence. A sensitivity analysis would strengthen the evaluation.
- Although Algorithm 1 is well-designed, the paper lacks runtime and complexity analysis. The scalability of the approach for large model pools (tens or hundreds of models) remains unclear.

**Questions:**

see above

---

> ### Author Response · Authors · 2025-11-22
> **Point-by-point responses to Reviewer NoyV**
>
> Thank you for acknowledging the novelty of our proposed method, the strength of our convex optimization formulation, and the breadth of our experimental evaluation. Please find our detailed responses below.
>
>
> - **The paper mentions potential mismatch between predicted and actual values but does not quantify its impact on constraint satisfaction or budget adherence....**
>
> **Response:**
>
> We thank the reviewer for asking this insightful question.
>
> While designing POLLINATOR, we acknowledged that performance and cost predictions can be wrong, and since the optimization problem uses them in objective and constraints, it will cause an optimality gap. The present work tackles it by breaking down the long-horizon into epochs, so that at the end of each epoch, the feedback (actual performance yield and cost incurred) from that epoch can be incorporated into the primal optimization problem (Eq. 1) – giving it a chance to course-correct.
>
> We note that a couple of recent works accepted at NeurIPS ‘25 offers a theoretical analysis: a) MESS+ [1], disclosed in late-October ‘25, proves a bound on the number of constraint violations; b) PORT [2], also disclosed in late-October ‘25, guarantees competitive ratio of online serving, after making certain assumptions on the efficacy of the predictors. We believe the theoretical analyses assume restricted settings which the present work doesn‘t consider. However, we leave a thorough and careful analysis of optimality gap/competitive ratio/constraint violation for future work.
>
>
> We have added this clarification in **Appendix A.1** of the revised PDF.
>
>
> [1] Woisetschläger, Herbert, et al. "MESS+: Dynamically Learned Inference-Time LLM Routing in Model Zoos with Service Level Guarantees.”
>
> [2] Wu, Fangzhou, and Sandeep Silwal. "PORT: Efficient Training-Free Online Routing for High-Volume Multi-LLM Serving.”
>
>
> - **Although Algorithm 1 is well-designed, the paper lacks runtime and complexity analysis. The scalability of the approach for large model pools (tens or hundreds of models) remains unclear.**
>
> **Response:**
>
> **Computational Complexity:**
> The complexity of Algorithm 1 is dominated by sorting, as detailed in Line 4. When the model pool size is $N$, the naive sorting takes $\mathcal{O}(N \log N)$. Rest of the operations are linear-time, $\mathcal{O}(N)$, as detailed below:-
> - **Predictor Invocation (Line 3).** Fetching the ex-ante performance and cost predictions takes $\mathcal{O}(N)$ time.
> - **Utility Computation (Line 4).** Calculating the utilities $\{u_{ij}\}_{j=1}^{N}$ also requires $\mathcal{O}(N)$ time – see Sec. 2.2.
> - **Utility Sorting (Line 4).** Sorting the computed utilities is the most expensive step, dominating the overall runtime with a complexity of $\mathcal{O}(N \log N)$.
> - **Iterative Thresholding (Lines 6-12).** The loop, in the worst case, may scan all utilities, taking $\mathcal{O}(N)$ time.
> - **Primal Serving Scheme (Line 14).** Constructing the primal serving scheme involves an additional $\mathcal{O}(N)$ pass.
>
> We have included the complexity analysis in **Appendix A.8** of revised PDF.
>
> **Wallclock Runtime:**
> As detailed in **Table 7** in the **Appendix A.1**, p99 latency incurred by Algorithm 1 is ~80ms across the range of $14$ datasets.
>
> **Handling Large Model Pool:**
> In practice, few commercial prompt routers are deployed with model pools of the size ~100 (HuggingChat Omni routes across 115 models [3]). More often, they route within the same model family (due to considerations arising from lack of prompt portability – what works best with GPT needn’t work with Gemini, as seen in their guides), thus limiting the model pool size to ~10.
>
> However, in the hypothetical scenario with ~10,000 models (which is practical only when routing across LoRAs, as done by vLLM Semantic Router), we would embrace a two-stage design a la recommender system. The first stage (a high-recall one) would cut down the pool to ~100, which the second, high-precision stage will evaluate as per Algorithm 1.
>
> We have added this clarification in **Appendix A.1** of the revised PDF.
>
> [3] https://news.ycombinator.com/item?id=45623284

---

> > ### Author Response · Authors · 2025-11-25
> > **Looking forward to your feedback**
> >
> > Dear Reviewer,
> >
> > Thank you for your valuable feedback and constructive comments.
> >
> > In our rebuttal, we have included additional experiments and enhanced explanations, and we hope that we have addressed all the concerns you raised. We remain open to further discussion and would be happy to clarify any remaining questions.
> >
> > This is just a gentle reminder. We look forward to hearing from you at your convenience.
> >
> > Thank you.
> >
> > The Authors

---

> > > ### Comment · Reviewer_toZC · 2025-11-28
> > >
> > > Thank you for the authors' response. While it has addressed some of my concerns, I believe that stress tests under different levels of concurrency would more directly reflect the system's performance.

---

> > > > ### Author Response · Authors · 2025-12-03
> > > > **Stress-Test Experiment Updated in Reviewer toZC Thread**
> > > >
> > > > Perhaps by mistake,  the reviewer **toZC** replied in another reviewer’s (**NoyV**) thread. However, we have addressed the requested experiment and provided the additional stress-test results (**Figure 4, Tables 8,9 in Appendix A.1** in revised PDF) under the correct thread for reviewer **toZC**.

---

### Official Review · Reviewer_hrhM · 2025-11-01

**Soundness:** 2
**Presentation:** 2
**Contribution:** 3
**Rating:** 4
**Confidence:** 3

**Summary:**

This paper presents a novel router designed to select the optimal LLM producer for a request, balancing cost and performance in real-time. The system integrates a frugal predictor that combines graph-based semi-supervised learning with an Item Response Theory head. This is coupled with an online dual-based optimizer that formulates the matchmaking as a strongly convex problem for efficient, real-time serving. Experiments demonstrate superior cost-performance tradeoffs, achieving performance gains at a fraction of the cost  across various in-domain and out-of-domain benchmarks.

**Strengths:**

- The paper proposes a novel approach that leverages GCN with an IRT head, reducing the training cost significantly.
- The dual serving scheme allows online optimization and offers better cost-performance tradeoff.
- Empirical results show that the method achieves superior performance gain at a lower cost.

**Weaknesses:**

- It's unclear how the GCN helps improve the router. The ablation study does not show any particular trend on the graph neighbour size (tab. 4), questioning its role. What if we remove the GCN part (similar to size 1), or simply apply k NN averaging?
- The experimental setting is not presented well. How do you create the Performance-First/Cost-First/Balance setting? The paper should include a thorough comparison across the whole spectrum of performance and cost (see Figure 4 in RouterBench for example). Also, the oracle result where we choose the best and cheapest LLM for each sample should be reported for reference.
- The results look strange. How can we achieve lower costs than the cheapest model (Tab. 1 Cost-First, Tab. 2 Balanced/Cost-First, and Tab. 3)?

**Questions:**

Please see weaknesses.

---

> ### Author Response · Authors · 2025-11-22
> **Point-by-point responses to Reviewer hrhM (1/2)**
>
> Thank you for carefully reviewing our submission and highlighting the strengths and novelty of our paper. Please find below our responses to the specific comments you made:-
>
>
> - **It's unclear how the GCN helps improve the router.......What if we remove the GCN part (similar to size 1), or simply apply k NN averaging?**
>
> **Response:** We thank the reviewer for the insightful question. To directly address the question of whether the GCN contributes meaningfully to the router, we performed the suggested ablation where we remove the GCN module entirely and replace it with **simple kNN averaging** (i.e., non-learned neighbor aggregation without message passing).
>
>
> **Table:** Impact of GCN Module: removing the GCN lowers performance and raises cost.
>
> | **Method** | **Datasets** |  |  |  |  |  |  |  |
> |-----------|--------------|--|--|--|--|--|--|--|
> |           | **In-Domain** |  | **Out-of-Domain** |  | **MMLU-Pro** |  | **BFCL-V3 (ToolCall)** |  |
> |           | Perf (%)↑ | Cost ($)↓ | Perf (%)↑ | Cost ($)↓ | Perf (%)↑ | Cost ($)↓ | Perf (%)↑ | Cost ($)↓ |
> ||||||||||
> | **POLLINATOR w/o GCN** | 80.20 | 0.63 | 79.18 | **0.13** | 74.23 | 2.55 | 86.67 | 0.02 |
> | **POLLINATOR** | **81.38** | **0.39** | **87.37** | 0.14 | **79.18** | **0.88** | **92.00** | **0.006** |
>
>
> As shown in the above Table, the kNN-only variant  (POLLINATOR w/o GCN) performs consistently worse than our proposed model (POLLINATOR) across all datasets:
>
> - **In-Domain:** performance drops from 81.38% → 80.20%, with cost increasing 0.39 → 0.63.
> - **Out-of-Domain:** significant performance degradation, 87.37% → 79.18%.
> - **MMLU-Pro:** performance drops 79.18% → 74.23%, and cost increases sharply 0.88 → 2.55.
> - **BFCL-V3:** performance decreases 92.00% → 86.67% with cost increasing 0.006  → 0.02
>
> These results demonstrate that simple kNN averaging cannot substitute for the learned message-passing performed by the GCN. While the neighborhood-size ablation in Table 4 does not show a strictly monotonic trend with respect to kkk, adjusting kkk only changes the number of neighbors and does not replicate the GCN’s learned aggregation. The kNN-only ablation clearly shows that removing the GCN consistently degrades performance, confirming its essential role in our router design.
>
> We have added this experiment to **Section 5** (Ablation Study: Impact of GCN Module), with the corresponding results reported in **Table 5** of the revised manuscript.
>
> -  **The experimental setting is not presented well. How do you create the Performance-First/Cost-First/Balance setting? The paper should include a thorough comparison across the whole spectrum of performance and cost ...  Also, the oracle result where we choose the best and cheapest LLM for each sample should be reported for reference.**
>
> **Response:**
>
> In order to highlight a few salient points on the cost-quality Pareto, we coin performance-first, balanced and cost-first configurations. In particular, in the performance-first setting, we set a large $C$ in Equation 1, so that the optimizer has a large room to maximize accuracy (performance). In cost-first, we set $C$ in a stringent manner, thus yielding lower performance. Balanced setting sets $C$ to a medium value.
>
> We added this clarification in **Section 3.2** within the  experimental set-up.
>
> **Performance-Cost Spectrum:**
> We have now included the entire spectrum of performance and cost across all datasets along with the corresponding oracle performance, into **Figure 1** in **Section 4.1** of the revised PDF.
>
> In the following Table, we have shown the **oracle results**, where for each request, we select the best-performing and lowest-cost LLM available, assuming we are allowed to broadcast the request to all the models, and can evaluate the ensuing responses. For comparison, we also include the performance of our proposed router (POLLINATOR) as well as the closest baseline, the MIRT router. These oracle upper bounds are  provided in **Table 11** of **Appendix A.2** of the revised PDF.
>
> **Table:** Oracle performance across all datasets
> | **Method**     | **In-Domain**        |               | **Out-of-Domain**     |               | **MMLU-Pro**          |               | **BFCL-V3 (ToolCall)** |               |
> |----------------|-----------------------|---------------|------------------------|---------------|------------------------|---------------|--------------------------|---------------|
> |                | Perf (%)↑             | Cost ($)↓     | Perf (%)↑              | Cost ($)↓     | Perf (%)↑              | Cost ($)↓     | Perf (%)↑                | Cost ($)↓     |
> | **Oracle**     | 95.52 | 0.19 | 98.32  | 0.03 | 98.17   | 0.79  | 100.0  | 0.01 |
> | **MIRT-Router**| 80.67 | 0.42  | 87.12  | 0.14  | 78.84  | 1.18 | 90.66  | 0.008   |
> | **POLLINATOR** | 81.38  | 0.39  | 87.37  | 0.14      | 79.18    | 0.88  | 92.00 | 0.006 |

---

> ### Author Response · Authors · 2025-11-22
> **Point-by-point responses to Reviewer hrhM (2/2)**
>
> -  **The results look strange. How can we achieve lower costs than the cheapest model (Tab. 1 Cost-First, Tab. 2 Balanced/Cost-First, and Tab. 3)?**
>
> **Response:**   Thank you for pointing this out. The confusion comes from how the **small LLM baseline** is  specified in the baseline **IRT-Router** [1] paper. The authors designate **Ministral-8B-Instruct-2410** as the *“small LLM”* (Section 5.3 of [1]), likely based on its parameter size, not its cost.
>
> For consistency, we use their reported performance and cost values directly. To clarify the misunderstanding, we list below the top-3 cheapest LLMs for each dataset. The complete performance–cost tables of all LLMs are included in **Appendix A.5 (Tables 13–16)** of the revised PDF.
>
> ---
>
> **In-Domain: Top-3 Cheapest LLMs**
>
> | Model | Performance | Cost ($) |
> |--------------------------|-------------|----------|
> | Qwen2.5-7B-Instruct | 0.6138 | 0.0607 |
> | Llama3.1-8B-Instruct | 0.3186 | 0.0877 |
> | Mistral-7B-Instruct-v0.2 | 0.1042 | 0.1123 |
>
> **Note:** POLLINATOR’s cost-first value (0.26 in Table 1) is *not* lower than the cheapest models — it is lower only relative to the IRT-Router’s chosen “small LLM.”
>
>
>  **Out-of-Domain: Top-3 Cheapest LLMs**
>
> | Model | Performance | Cost ($) |
> |--------------------------|-------------|----------|
> | Qwen2.5-7B-Instruct | 0.5994 | 0.0147 |
> | Llama3.1-8B-Instruct | 0.4462 | 0.0256 |
> | Mistral-7B-Instruct-v0.2 | 0.1215 | 0.0344 |
> **Note:** POLLINATOR’s balanced and cost-first costs (0.10 and 0.09) reported in Table 2, are both **higher** than these cheapest LLMs.
>
>
> **BFCL-V3 (ToolCall): Top-3 Cheapest LLMs**
> | Model | Performance | Cost ($) |
> |-------------------|-------------|----------|
> | Gemini-1.5-Flash | 0.8800 | 0.0049 |
> | GPT-4.1-Nano | 0.7733 | 0.0059 |
> | GPT-4o-Mini | 0.9067 | 0.0088 |
>
>
>
> **Note:**  In the BFCL-V3 ToolCall (Table 3), we designate  **GPT-4.1-Nano** as the small LLM (see Page 6, footnote). GPT-4.1-Nano (our designated small LLM) is **not** the cheapest; Gemini-1.5-Flash remains lower-cost than our router cost (0.006).
>
> **MMLU-Pro: Top-3 Cheapest LLMs**
> | Model                                   | Performance (%) | Cost ($) |
> |-----------------------------------------|------------------|----------|
> | Gemini-1.5-Flash-002                    | 63.80            | 0.3316   |
> | Gemini-2.0-Flash-Exp                    | 78.10            | 0.8765   |
> | GPT-4.1-Nano                            | 61.68            | 0.8847   |
>
>
>  **MMLU-Pro:**  For MMLU-Pro dataset, we selected Meta-Llama-3.1-70B as the small LLM (see Page 6, footnote).The selected small LLM, Llama-3.1-70B, is also *not* the cheapest model. The lowest-cost model in this pool is **Gemini-1.5-Flash (0.33)**.
>
> We have added this clarification in Section 3.2 (Baselines) and included the full performance–cost tables in **Appendix A.5 (Tables 13–16)** of the revised PDF.
>
> [1]  Wei Song et al. IRT-router: Effective and interpretable multi-LLM routing via item response theory. 2025

---

> > ### Author Response · Authors · 2025-11-25
> > **Looking forward to your feedback**
> >
> > Dear Reviewer,
> >
> > Thank you for your valuable feedback and constructive comments.
> >
> > In our rebuttal, we have included additional experiments and enhanced explanations, and we hope that we have addressed all the concerns you raised. We remain open to further discussion and would be happy to clarify any remaining questions.
> >
> > This is just a gentle reminder. We look forward to hearing from you at your convenience.
> >
> > Thank you.
> >
> > The Authors

---

### Author Response · Authors · 2025-11-22
**Summary of Revised Manuscript**

We thank the reviewers for their constructive feedback. In response, we have updated the manuscript with substantial clarifications, additional experiments, and revised analyses.  All updates are highlighted in **blue** in the revised PDF.


## ---- Summary of Revisions in the Updated Manuscript---


## 1. New Experiments and Ablations
- **GCN removal ablation:** Added a new experiment replacing the GCN with simple kNN averaging. The newly added paragraph “Impact of the GCN Module” (**Section 5**, Ablation Study) shows that replacing the GCN with simple kNN averaging leads to consistent performance degradation with increased cost (see **Table 5**)
- **Oracle upper-bound results:** Added  oracle performance to establish routing upper bounds (**Appendix A.2**, **Table 11**).
- **Full performance–cost spectrum:** Added complete performance-cost spectrum for all datasets, including oracle references (**Figure 1**, **Section 4.1**).


## 2. Clarified Modeling and Theoretical Aspects
- **Clarified Configuration Definitions**
  The definitions of the *performance-first*, *balanced*, and *cost-first* configurations have been clarified in **Section 3.2** (Metrics & Baselines)
- **Cost modeling:** Expanded explanation of real-world pricing factors (provider, region, reservation type, negotiated rates, PTU) and justified abstraction via per-million-token rate cards. Clarified how the optimizer is re-solved upon pricing changes (**Section 2.2**).
- **Constraint modeling:** Detailed how linear constraints (e.g., minimum-volume commitments, p99 latency) integrate naturally into the Lagrangian formulation (**Section 2.2**).
- **Theoretical context:** Added discussion of recent NeurIPS’25 analyses (MESS+, PORT) on constraint violations and competitive ratios (Appendix A.1).

## 3. System  Computational Complexity, Scalability and Robustness
We have added a detailed Scalability and Robustness section in **Appendix A.1**, analyzing POLLINATOR’s end-to-end latency, stress testing under varying levels of concurrency throughput, and robustness under dynamic pricing and provider availability. To support this analysis, we also added **Table 7**, which reports per-query inference times across all datasets.
- **Prediction–optimization mismatch:** Clarified that the optimization is re-solved each epoch, updating dual variables to correct prediction errors (**Appendix A.1**).
- **Computational complexity:** Added full runtime breakdown establishing **$\mathcal{O}(N \log N)$** complexity for Algorithm 1 (**Appendix A.8**).
- **Latency:** Reported end-to-end latency across all datasets in **Table 7** (**Appendix A.1**).
- **Stress Test :**  Added realistic load-test result under varying levels of concurrency in **Figure 4** and **Tables 8,9** (**Appendix A.1**)

- **Scalability:** Added discussion on handling large model pools, nearest-neighbour lookup efficiency, throughput with vector DBs (e.g., Qdrant), and scaling under asynchronous execution (**Appendix A.1**).



## 4. Clarifications on Overhead, Drift, and Provider Availability
- **Routing overhead:** Clarified that the router invokes exactly one LLM per request, with no additional token overhead beyond a lightweight lookup (**Appendix A.1**).
- **Handling drift:** Added explanation of robustness to LLM ability drift and query-distribution drift via predictor/optimizer retraining, supported by strong OOD results (**Table 2**) and an additional data-drift experiment presented in **Table 10** in **Appendix A.1**.
- **Dynamic pricing & provider availability:** Clarified how pricing updates trigger re-solving of the primal problem, and added fallback routing strategy for provider downtime (**Appendix A.1**).


## 5. Additional Improvements
  The definition of the Small LLM baseline has been clarified, and the exact models used for each dataset are now specified following Song et al. (2025) in **Section 3.2**. Added full performance–cost tables in **Appendix A.5** (**Tables 13–16**).

---

### Author Response · Authors · 2025-12-03
**Summary of Reviewer Feedback and Response**

We thank the reviewers for their sincere comments.

---

### Highlighted Strengths:-

The reviewers were unanimous in highlighting certain strengths:

- **(a)** Timeliness and **practical importance** of the problem formulation
- **(b)** **Novel** performance predictor architecture that integrates the cost-effectiveness of **GNN** with the interpretability of **IRT**
- **(c)** Principled and theoretically well-founded treatment of decision-making via **convex optimization**
- **(d)** Strong empirical performance demonstrated across a comprehensive array of **14 datasets**

---

### Actionable Feedback Summary:-

The actionable feedback can be broadly classified into the following buckets:

- **(a)** Seeking additional clarifications
- **(b)** Seeking additional experiments — *e.g., ablations, robustness to drift, etc.*
- **(c)** Seeking performance characteristics of the inference system — *e.g., load-test, latency, scalability*
- **(d)** Seeking clarification on the assumptions and formulation — *e.g., dynamic and preferential pricing, etc.*

---
Please refer to the revised manuscript (PDF) for the detailed updates summarized below.
Below, we provide a crisp summary (along with pointers) of how we addressed _all_ the actionable comments, even when we felt they lie outside the purview of the present work:

---

# A Summary of Conversation Dynamics


| **No.** | **Reviewer**| **Comment** | **Response**  | **Follow-up Comment** | **Follow-up Response** |
|-|-|-|-|-|-|
| 1 | `hrhM`  | ✅What if we remove GCN or simply apply kNN averaging?"   | Ablation in **Table 5 (Sec. 5)**. | | |
| 2 | `hrhM` | ✅Clarify Perf-First/Cost-First/Balance setting?    |  Clarified  in **Sec. 3.2.** | | |
| 3 | `hrhM`  | ✅ Include the whole spectrum of performance and cost. Report the oracle result . | Added full spectrum in **Fig. 1 (Sec. 4.1)**; Added Oracle results in **Table 11 (App. A.2)**.  | | |
| 4 | `hrhM` | ✅How can we achieve lower costs than the cheapest model ?    | The cheapest model isn't always the smallest model. Added clarification in **Sec. 3.2**, and, added the rate-card in **Tables 13-16 (App. A.2)**.| | |
|||||||
| 1 | `NoyV` | ✅Quantify prediction–actual mismatch impact on constraints/budget.               |  Added discussion **(App. A.1)** .| | |
| 2 | `NoyV` | ✅Although Algorithm 1 is well-designed, the paper lacks runtime and complexity analysis. The scalability of the approach for large model pools remains unclear.  |   Added complexity analysis in **App A.8**;  Provided end-to-end latency in **App. A.1**; Clarified handling of large model pools in **App. A.1**| | |
|||||||
| 1 | `ex1Y` | ✅Oversimplified cost/constraints; industrial constraints missed.              |  Clarified cost and constraint modelling in **Sec 2.2.**   | | |
| 2 | `ex1Y` | ✅Comparison with RL routing;static LLM ability assumption; adaptation to drift?.   |  See rebuttal responses. Clarified drift handling; results in **App. A.1** & **Table 10**. |  | |
| 3 | `ex1Y` | ✅Is the proposed method sensitive to the hyper parameter selection ?     |  Pointed out Sensitivity  ablation (**App. A.9, Table 21**) to showcase the robustness . |  | |
| 4 | `ex1Y` | ✅ What are some alternative methods for cost optimization in LLM deployment?   |  Detailed in the rebuttal response. | | |
|||||||
| 1 | `toZC` | ✅How is latency/token overhead managed when invoking multiple models?            |  Detailed in the rebuttal response. | | |
| 2 | `toZC` | ✅Mitigation of prediction–optimization mismatch? Calibration/uncertainty methods?                      |  Calibration is integrated. See rebuttal for details.| | |
| 3  | `toZC`  |✅ Can the authors provide additional experiments or analysis to evaluate scalability, particularly in terms of end-to-end latency, system throughput, and robustness under dynamic model pricing or service availability changes?  |  Added new section scalability and robustness of POLLINATOR with all the results and discussion (**App A.1**)  |  ✅ Stress tests under different levels of concurrency would more reflect the system's performance.   |  ✅ We conducted a realistic load-test under varying levels of concurrency  (**Figure 4, Tables 8,9 in App A.1**) |

---

### Meta-Review · Area_Chair_DycK · 2026-01-10

**Summary:**

Major concerns from the reviewers:

- It's unclear how the GCN helps improve the router. The ablation study does not show any particular trend on the graph neighbour size (Tab. 4).

- The experimental setting is not presented well.

- The results look strange. How can we achieve lower costs than the cheapest model (Tab. 1 Cost-First, Tab. 2 Balanced/Cost-First, and Tab. 3)?

- The paper mentions a potential mismatch between predicted and actual values but does not quantify its impact on constraint satisfaction or budget adherence. A sensitivity analysis is required.

- The paper lacks runtime and complexity analysis. The scalability of the approach for large model pools remains unclear.

- The modeling comes with oversimplified costs and constraints. The paper did not take into account other practical factors, such as hardware deployment overhead or regional pricing differences. Other common industrial constraints are not considered in the proposed approach, e.g., minimum LLM usage volume commitments.

- The paper lacks in-depth theoretical analysis and comparisons with RL-based dynamic routing methods. Also, it only assumes static LLM abilities.

- Is the proposed method sensitive to the hyperparameter selection?

- What are some alternative methods for cost optimization in LLM deployment?

- The main limitation of the paper lies in the practicality and scalability of its proposed elastic model routing mechanism. The paper lacks a concrete analysis of how such routing can be implemented efficiently in real deployment scenarios.

- The framework’s predictive component relies heavily on estimated performance and cost metrics without a clear strategy for handling uncertainty or prediction errors.

- Operational constraints such as service rate limits, model drift, or dynamic pricing are not modeled or evaluated, raising concerns about the system’s stability and reliability.

**Reviewer Concerns:**

The authors have addressed many of the reviewers' concerns with additional experiments and analysis. The authors have also incorporated these changes into the revised paper. However, several points remain not well addressed in the rebuttal.

For Reviewer hrhM's concern, the reviewer asked what happens if we remove the GCN part (similar to size 1) or use simple k‑NN averaging. The rebuttal provides results for the k‑NN variant, but it would be stronger to also show the "size‑1 / no‑GCN" result explicitly.
In addition, in the k‑NN study, it seems unclear in the response what value of $k$ is used, and whether varying $k$ changes the results. In the reported numbers, the k‑NN variant sometimes appears to have a lower cost, which would benefit from an explicit explanation. Regarding the value of $C$ for the performance‑first / cost‑first / balanced settings, the rebuttal only states that $C$ is large, stringent, or medium, without providing concrete values. Thus the reviewer's concern about the clarity of the experimental setup is not fully resolved. Moreover, in Table 1 in the paper, the results of POLLINATOR for performance-first and balanced settings are the same, which is interesting. A brief explanation here would also be helpful.

For Reviewer NoyV's concern that "the paper mentions potential mismatch between predicted and actual values but does not quantify its impact on constraint satisfaction or budget adherence. A sensitivity analysis would strengthen the evaluation", although the rebuttal gives a discussion about this issue, a small empirical or simple theoretical sensitivity analysis would be much better. Deferring this to future work does not fully resolve the reviewer’s concern, so this issue remains only partially addressed.

For Reviewer ex1Y's concern that "other common industrial constraints are not considered in the proposed approach, e.g., minimum LLM usage volume commitments", the rebuttal explains that the proposed formulation can generally handle linear constraints and strongly convex objective. The response is somewhat general. It would be more convincing to present more details about how it can be concretely instantiated for the proposed problem by this reviewer.

**Reviewer Scores:**

According to the discussions above, since some concerns remain insufficiently resolved, it is reasonable to expect that Reviewer hrhM and Reviewer ex1Y may keep their score of 4 unchanged.

---

### Decision · Program_Chairs · 2026-01-26

Reject